# Motor generated torque drives coupled yawing and orbital rotations of kinesin coated gold nanorods

Mitsuhiro Sugawa [1,2✉], Yohei Maruyama[1], Masahiko Yamagishi [1], Robert A. Cross [3] & Junichiro Yajima [1,2,4✉]

Kinesin motor domains generate impulses of force and movement that have both translational and rotational (torque) components. Here, we ask how the torque component influences function in cargo-attached teams of weakly processive kinesins. Using an assay in which kinesin-coated gold nanorods (kinesin-GNRs) translocate on suspended microtubules, we show that for both single-headed KIF1A and dimeric ZEN-4, the intensities of polarized light scattered by the kinesin-GNRs in two orthogonal directions periodically oscillate as the GNRs crawl towards microtubule plus ends, indicating that translocating kinesin-GNRs unidirectionally rotate about their short (yaw) axes whilst following an overall left-handed helical orbit around the microtubule axis. For orientations of the GNR that generate a signal, the period of this short axis rotation corresponds to two periods of the overall helical trajectory. Torque force thus drives both rolling and yawing of near-spherical cargoes carrying rigidly-attached weakly processive kinesins, with possible relevance to intracellular transport.

[1] Department of Life Sciences, Graduate School of Arts and Sciences, The University of Tokyo, 3-8-1, Komaba, Meguro-ku, Tokyo 153-8902, Japan. [2] Komaba Institute for Science, The University of Tokyo, 3-8-1, Komaba, Meguro-ku, Tokyo 153-8902, Japan. [3] Centre for Mechanochemical Cell Biology and Division of Biomedical Sciences, Warwick Medical School, University of Warwick, Coventry CV4 7AL, UK. [4] Research Center for Complex Systems Biology, The University of Tokyo, 3-8-1, Komaba, Meguro-ku, Tokyo 153-8902, Japan. ✉email: mitsuhiro.sugawa@bio.c.u-tokyo.ac.jp; yajima@bio.c.u-tokyo.ac.jp

Kinesins are adenosine 5'-triphosphate (ATP)-driven and microtubule-based molecular motors with multiple mechanical functions, including cargo transport along microtubules, the regulation of microtubule dynamics, and the sliding of bundled microtubules[1–3]. Recent advances in techniques for 3D particle tracking and simultaneous measurements of force and torque have revealed that kinesins have translational and rotational degrees of freedom on microtubules, such that their motilities consist of both helical[4–8] and rotational[9] motions. Single-molecule kinesin-1 dimers, which are highly processive and accurately track single microtubule protofilaments, unidirectionally rotate their stalk domain as they move processively via a hand-over-hand mechanism, especially at high ATP concentration[9]. Mutagenic elongation of the neck linker of kinesin-1 dimers increases the probability of protofilament switching during stepping along the microtubule[10], so that the neck-linker mutant dimers retain processivity but switch protofilaments more readily, driving left-handed helical motion of multiple motor-coated beads around the axis of the microtubule[6]. These studies suggest that the protofilament-tracking ability of wild type kinesin-1 dimers may derive from their rotary hand-over-hand mechanism[9] and their neck stability[6,8]. On the other hand, some kinesins do not follow the microtubule protofilament axis. Wild-type kinesin-2 and -8, which have longer neck-linker domains than kinesin-1, and kinesin-6, which has an atypical neck region, do not track protofilaments. All these kinesins move towards microtubule plus ends along left-handed helical trajectories, as shown in a bead assay[6,11], a microtubule surface-gliding assay[8,11–13] and a single-molecule tracking assay[8]. Furthermore, teams of non-processive dimeric kinesin-14 Ncd[14–16], which moves towards microtubule minus ends, as well as teams of truncated single-headed kinesins-1[17,18], -3[19], -5[4,20], and -6[11] motors have all been shown to drive corkscrewing motions of microtubules in surface-gliding assays. The handedness of these motions is consistent. Whilst plus-end directed kinesins move along a left-handed helix, minus-ended kinesins move along a right-handed helix. These helical motilities imply that the stroke (the unitary impulse of force and movement generated by the motor domains) has an off-axis component, driven either by conformational changes in the motor domain or by the directionally biased selection of microtubule binding sites (or potentially both)[6,8,11,15,17,19]. The underlying mechanisms and functional relevance of the torque component within teams of non-processive kinesins remain elusive.

To gain further insight, we have developed a novel, to our knowledge, motility assay in which gold nanorods (GNRs), coated with a team of kinesin molecules, move unhindered along suspended microtubules. Our assay uses light scattered by the kinesin-coated GNRs to track their position, and the polarization of the scattered light to report their orientation. We performed the GNR motility assay on the plus-end-directed kinesins KIF1A, which is monomeric, and ZEN-4, which is a weakly processive dimer.

## Results

### GNR imaging for particle tracking and polarization measurements.
We developed a motility assay based on tracking of kinesin-coated GNRs. Focusing of light scattered by the GNRs allows positional tracking, whilst the ratio of the polarization of this scattered light in orthogonal directions reports the orientation of the GNR as it moves over the microtubule surface (Fig. 1a and Materials and Methods). GNR images were obtained by laser dark-field imaging using highly inclined illumination giving almost total internal reflection[21,22] via a perforated dichroic mirror[23]. The position of each GNR spot was measured in three

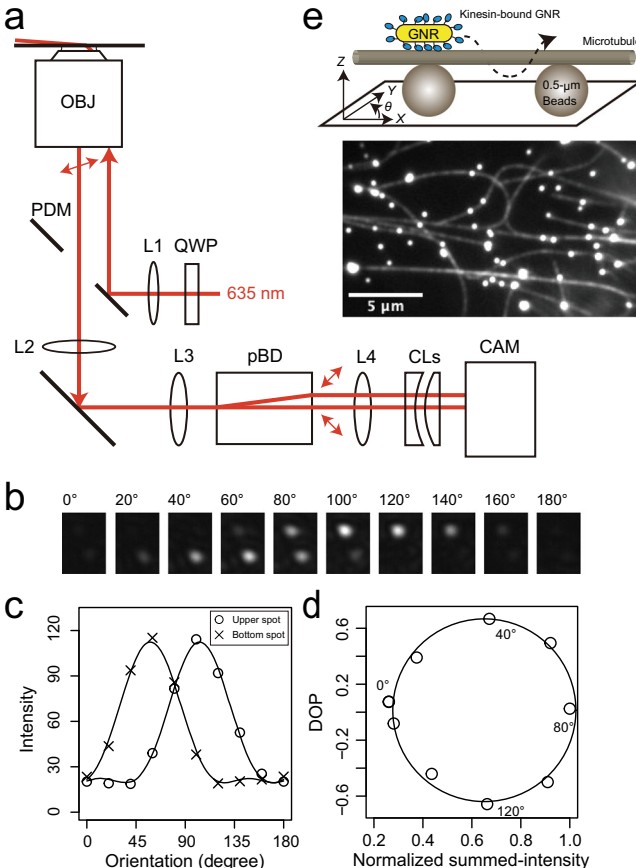

**Fig. 1 GNR motility assay. a** Schematic drawing of the optics for the gold nanorod (GNR) motility assay. OBJ objective lens, PDM perforated dichroic mirror, QWP quadruple wave plate, pBD polarizing beam displacer, CL cylindrical lens, CAM CMOS camera, L lens. **b–d** Calibration for polarization measurements. **b** Series of pairs of scattered-light spots of microbeads (0.1 μm) modulated by a polarizer, which was located in front of L3, for emulating GNR spots with various orientations in the plane. **c** Intensity profiles of a pair of scattered-light spots ($I_{bottom}$, $I_{upper}$) shown in the upper images. The plots of $I_{bottom}$ and $I_{upper}$ were fitted by intensity functions of the GNR orientation (Materials and Methods). **d** Degree of polarization (DOP), ($I_{bottom} - I_{upper}$)/($I_{bottom} + I_{upper}$) as a function of the normalized summed-intensity of two scattered-light spots, ($I_{bottom} + I_{upper}$). **e** Schematic drawing of the GNR motility assay in which a kinesin-coated GNR moves along a microtubule suspended with fluorescent microbeads (0.5 μm) attached to a glass substrate. Fluorescence image of Cy5-labeled microtubules suspended on fluorescent microbeads. The bright spots are the fluorescent microbeads.

dimensions using a cylindrical lens[24] (Supplementary Fig. 1). Polarization measurements of GNRs were performed using a combination of the polarized incident laser beam and a polarizing beam displacer, which projected two orthogonal polarization signals on to the camera, allowing us to measure 0° to 180° GNR orientations in the sample plane (Fig. 1b–d and Materials and Methods). Kinesin constructs were prepared with a biotinylated peptide (Avi-tag) fused to their C-terminus. Biotinylated GNRs (40 × 68 nm) were coated in streptavidin and biotinylated kinesin molecules were attached to this streptavidin layer (Materials and Methods). We used GNRs with relative low aspect ratio in order to detect rotations of kinesin coated GNR (kinesin-GNR) on the microtubule. A GNR with a higher aspect ratio may inhibit rotations of a kinesin-GNR, since the situation would then approach the geometry of two sliding microtubules, which are

known to maintain their axes parallel[16]. We sought to saturate kinesin attachment to the GNRs, and we estimate the maximum capacity of kinesin molecules attached to each GNR to be about 100 molecules (surface area of GNR/occupied area of kinesin ≈ $1.1 \times 10^4$ nm$^2$/10$^2$ nm$^2$) with about one-fourth of the attached kinesin molecules able to engage with the microtubule and generate driving force and torque at any one time. Assuming that the duty ratio of weakly processive kinesin is about 20%, an average of 5 kinesin motor domains might generate force at any one time, thereby allowing the kinesin-GNR to move processively along the microtubule. Microtubules were linked to microbeads (0.5 μm) via the antigen–antibody system for β-tubulin, and the microtubules suspended so that kinesin-coated GNRs were free to orbit the microtubules between bead-attachment points (Fig. 1e). Control experiments established that the $Z$-displacement of GNRs in the range of ±200 nm had a negligible effect on the GNR polarization measurement in our experimental system (Supplementary Fig. 2). Using the GNR motility assay, we performed a comparative analysis of two plus-end-directed kinesins, which have distinctive properties of processivity and oligomeric state: biased-diffusional motile and single-headed human kinesin-3 (KIF1A)[25] and weakly processive and dimeric *Caenorhabditis elegans* kinesin-6 (ZEN-4)[26].

**GNR motility assay for single-headed KIF1A.** We performed the GNR motility assay for single-headed KIF1A (1-366 amino acid residues (AAs))[25] at a saturating ATP concentration (2 mM ATP). GNRs carried multiple KIF1A molecules, allowing GNRs to move processively along microtubules. We found that the light-scattering intensities of each pair of KIF1A-coated GNR (KIF1A-GNR) spots, corresponding to orthogonal polarization signals from a single GNR, periodically oscillated in intensity during helical movement, with an increase in each signal corresponding to a decrease in the other (13 particles out of 62 tested KIF1A-GNRs) (Fig. 2a and Supplementary Movie 1). The degree of polarization (DOP) as a function of the summed-intensity of the two GNR spots depicted an elliptical trajectory, indicating continuous rotations of the polarization (Fig. 2b). The $X$-$Y$-$Z$-trajectory of the KIF1A-GNR showed steady left-handed helical motion around the microtubule long axis (Fig. 2c). Figure 2d shows that the time trajectories of the $X$- and $Y$-displacements, the DOP and summed-intensity, and the polarization angle as viewed from above the glass substrate. The time trajectory of the polarization angle exhibited continuous counterclockwise rotations during the left-handed helical motion of the KIF1A-GNR around the microtubule. The $Y$-displacement and the polarization signals of the KIF1A-GNR exhibited strong cross-correlation and periodicity (Fig. 2e), indicating that the GNR polarization were synchronized with the helical motion on the microtubule. We also observed clockwise and oscillatory rotations of the GNR polarization, synchronized with the helical motion of KIF1A-GNRs (Fig. 3a, b). The translational velocity (parallel to the microtubule axis) was 0.51 ± 0.12 μm s$^{-1}$ (mean ± standard deviation (SD), $n = 13$ GNRs), which is nearly equal to the microtubule sliding velocity of 0.49 μm s$^{-1}$ in the surface-gliding assay (Supplementary Fig. 3). The pitch of the helical motion measured in the $X$-$Y$-trajectory was 0.69 ± 0.36 μm (mean ± SD, $n = 33$ cycles in 13 KIF1A-GNRs), which is slightly shorter than that of the corkscrewing of microtubules on a surface of KIF1A of 0.89 ± 0.21 μm (mean ± SD, $n = 35$ cycles in 5 microtubules) ($P \ll 0.01$, Welch's $t$ test). The $X$-displacement in the microtubule axis during one periodic change in the GNR polarization signals (hereafter referred to as polarization pitch) was 0.69 ± 0.37 μm (mean ± SD, $n = 33$ cycles in 13 KIF1A-GNRs), which was almost equal to that of the helix pitch (Fig. 3c). We note that the measured helix

pitch is a combination of the helix pitch exerted by motors on the straight microtubule lattice and any supertwist pitch of the microtubule, with the supertwist depending on the number of the protofilaments. However, when the measured pitch is less than 1 μm, as it is here, the supertwist of the microtubule has negligible influence on the measured pitch (Supplementary Fig. 4 and Materials and Methods). We, therefore, report the uncorrected helix pitch unless otherwise mentioned.

**Biaxial rotations of kinesin-coated GNRs on microtubules.** The GNR motility assays of KIF1A revealed that the polarization of KIF1A-GNRs rotated periodically whilst moving along a helical trajectory toward microtubule plus ends. As a control to check whether rotation of the GNR polarization is coupled to helical orbiting of the supertwist of the microtubules or not, we studied GNRs coated with multiple kinesin-1 coated GNRs, which faithfully track protofilaments[6] rather than side-stepping. Polarization angle of the kinesin-1 coated-GNRs did not exhibit unidirectional rotation along a long-pitch helical path in our observation (Supplementary Fig. 5). This result indicates that when the orientation of a kinesin-coated GNR relative to the microtubule axis is constant during helical motion, unidirectional rotations of the GNR polarization are never observed (Fig. 4a and Supplementary Movie 2). Therefore, we hypothesized that unidirectional rotations of the GNR polarization are generated by rotations in three axes (rolling, yawing, pitching) of the kinesin-GNR during helical motion. We note that considering symmetry of gold nanorods and geometries of our polarization measurement, pitching and yawing of the GNR cannot be distinguished from each other in our polarization measurement. However, as shown below, the experimental results can be well explained by the combination of rolling and yawing in the kinesin-GNR. We constructed a model, in which a kinesin-coated GNR unidirectionally rotates about its yaw and roll axes during the short-pitch helical motion around the microtubule long axis (Supplementary Movie 3). Using the extrinsic rotations about the $Z$-, $Y$-, $X$-axes in that order, our proposed motion of the kinesin-coated GNR is given by

$$R_X(\omega_{Roll}t)R_Y(0)R_Z(\omega_{Yaw}t)p_0(\varphi) + helix(t, v, \omega_{Helix}), \quad (1)$$

where $R_X(\omega_{Roll}t)$ is the rolling matrix with the angular velocity $\omega_{Roll}$, $R_Y(0)$ the pitching matrix with zero angle, $R_Z(\omega_{Yaw}t)$ the yawing matrix with the angular velocity $\omega_{Yaw}$, $p_0$ an initial position of a kinesin-GNR with an initial phase angle $\varphi$, and $helix(t, v, \omega_{Helix})$ a helical displacement with translational velocity $v$ and angular velocity $\omega_{Helix}$ as functions of time $t$ (Materials and Methods). The initial phase angle $\varphi$ is defined by the relative angle to the microtubule axis when the kinesin-coated GNR is at the top position of its helical trajectory. The polarization angle measured in this study was the orientation of the GNR projected in the $X$-$Y$-plane. Thus, time trajectories of the GNR angle in the $X$-$Y$-plane are given by

$$\arctan[\cos(\omega_{Roll}t) \cdot \tan(\omega_{Yaw}t + \varphi)] \quad (2)$$

(Materials and Methods). To synchronize the rolling and the helix track, the angular velocity of the rolling ($\omega_{Roll}$) is identical to that of the helix track ($\omega_{Helix}$). And the angular velocity of the yawing ($\omega_{Yaw}$) is half that of the helix track because the GNR rotates 180° during one period of the helix track (Figs. 2d and 3a). Therefore, the time trajectories of the KIF1A-GNR angle were well explained by this model at $\omega_{Helix} = \omega_{Roll} = \pm 2\omega_{Yaw}$ (Fig. 4b). This means that a KIF1A-coated GNR unidirectionally rotates 180° about its short (yaw) axis in one period of short-pitch helical motion around the microtubule long axis, which is consistent with our observation that one full yawing motion is twice the

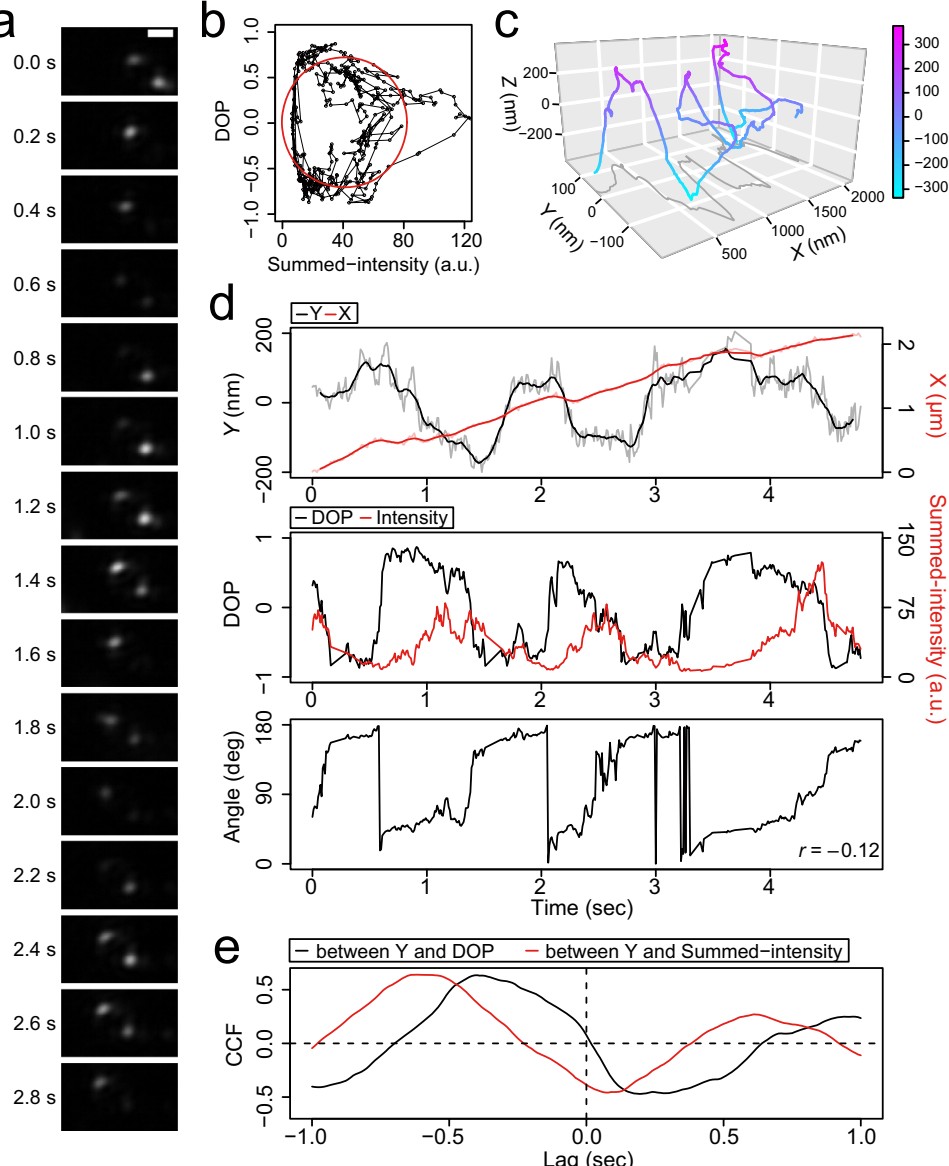

**Fig. 2 Typical data of the GNR motility assay for KIF1A. a** Montage of scattered-light images of one KIF1A-coated GNR (KIF1A-GNR) moving around a microtubule every 0.2 s in the presence of 2 mM ATP. The frame rate was 100 frames s⁻¹. Scale bar 2 μm. **b** Degree of polarization (DOP) as a function of the summed-intensity of the two KIF1A-GNR spots shown in **a**. The red line represents an ellipse fit. **c** 3D trajectory of the KIF1A-GNR through a moving-average filter of 15 frames. **d** The time trajectories of the X (red line)- and Y (black line)-displacements, the DOP (black line) and summed-intensity (red line) of the two spots, and the estimated angle of KIF1A-GNR shown in **a**. The value of r is Pearson's correlation coefficient between the Y-displacement and the angle. **e** Cross-correlation functions (CCFs) between the Y-displacement and the polarization signals.

rolling (helical) motion (Figs. 2d and 3c). The three types of rotations of the GNR polarization angle (counterclockwise, oscillatory, and clockwise rotations) can be reproduced by different initial phase angles ($\varphi$) of the model (Fig. 4b and Supplementary Movies 4 and 5). For example, rotations of the GNR polarization are counterclockwise at $\varphi = 0°$, oscillatory at $\varphi = 45°$ and 135°, and clockwise at $\varphi = 90°$. Since the initial phase angle $\varphi$ is a random variable in the assay, our model is consistent with experimental results in which these three types of polarization rotations of KIF1A-GNRs were observed.

In this model, the GNR yawing direction is also a determinant of apparent rotational patterns of the GNR polarization in the sample plane. Pearson's correlation coefficient between the Y-displacement and the angle of a kinesin-GNR is negative when it yaws clockwise and positive when counterclockwise (Fig. 4b and Supplementary

Fig. 6). Estimated by the correlation coefficient between the Y-displacement and the polarization angle, 8 out of 13 KIF1A-GNRs exhibited clockwise yawing whilst 5 out of 13 KIF1A-GNRs counterclockwise yawing (see also Figs. 2d and 3a, b). The directionalities of yawing of these kinesin-coated GNRs were not significantly different from an equal probability using binomial test ($P > 0.5$).

**GNR motility assay for dimeric ZEN-4.** We also performed the GNR motility assay for the weakly-processive dimeric kinesin, ZEN-4 (1-555 AAs)[26], again at a saturating ATP concentration (2 mM) (Materials and Methods). We again found that the polarization signals of the ZEN4-coated GNRs (ZEN4-GNR) periodically changed, correlated with the Y-displacements during steady forward movement (13 particles out of 49 tested ZEN4-GNRs) (Fig. 5a–c). The

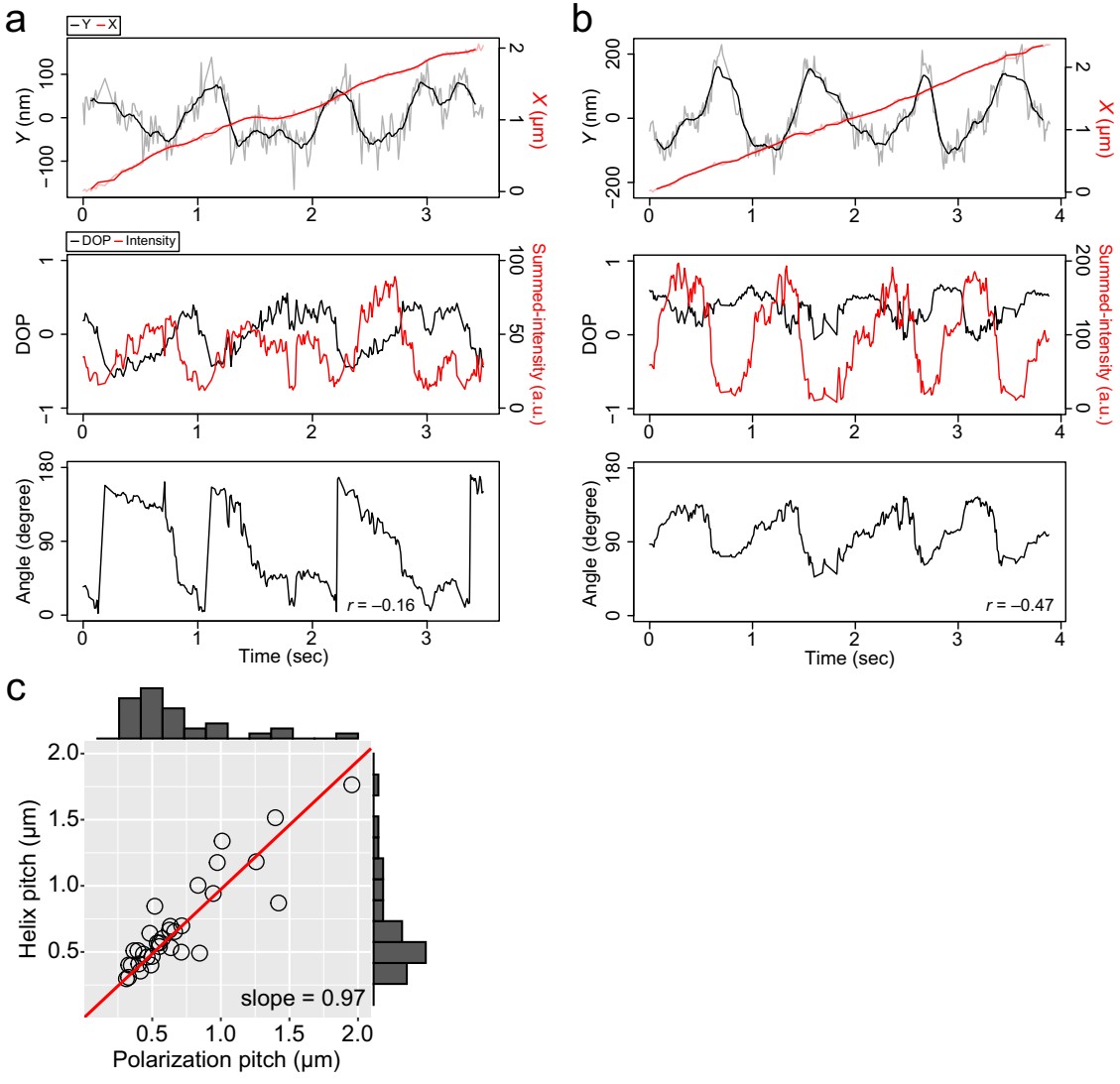

**Fig. 3 Coupled polarization rotation and helical motion of KIF1A-GNRs. a, b** The time trajectories of the $X$ (red line)- and $Y$ (black line)-displacements, the degree of polarization (DOP, black line) and summed-intensities (red line) of the two spots, and the estimated angles of KIF1A-GNRs. The time trajectories of the polarization in the image plane show clockwise rotation (**a**) and oscillatory rotation (**b**). The value of $r$ is Pearson's correlation coefficient between the $Y$-displacement and the angle. **c** Distribution of the helix and polarization pitches of KIF1A-GNRs. The helix and polarization pitches are the displacements during one periodic change in the $Y$-displacement and the polarization, respectively. The red line represents a linear fit ($y = ax$) of which the slope ($a$) is 0.97 ($R^2 = 0.95$).

analyses revealed the counter-clockwise, clockwise, and oscillatory rotations of these ZEN4-GNRs. The forward velocity of the ZEN4-GNR was $0.39 \pm 0.05\ \mu m\ s^{-1}$ (mean ± SD, $n = 13$ ZEN4-GNRs), which is comparable to the velocity in the bead assay[11]. The polarization pitch of the ZEN4-GNR was $0.67 \pm 0.20\ \mu m$ (mean ± SD, $n = 34$ cycles in 13 ZEN4-GNRs) in the microtubule axis, which is equal to the helix pitch of $0.67 \pm 0.21\ \mu m$ (mean ± SD, $n = 34$ cycles in 13 ZEN4-GNRs) (Fig. 5d). Thus, whilst ZEN4-GNRs moved more slowly than KIF1A-GNRs, the polarization rotation of ZEN4-GNRs was again phase-locked to their helical motion, as for KIF1A-GNRs. These results indicate that ZEN4-GNRs also unidirectionally rotates 180° about its short (yaw) axis in one period of short-pitch helical motion around the microtubule long axis. Estimated by the correlation coefficient between the $Y$-displacement and the polarization angle, 5 out of 13 ZEN4-GNRs exhibited clockwise yawing, whilst 8 out of 13 ZEN4-GNRs counterclockwise yawing. The directionalities of yawing of these ZEN4-GNRs were not significantly different from an equal probability using binomial test ($P > 0.5$).

**Unidirectional rotation of microtubules on single ZEN4-GNRs.** Our proposed model, in which kinesin-GNRs rotate around their yaw axes as they move along and around the microtubule long axis, predicts that immobilized kinesin-GNRs might drive rotation of an overlying short microtubule about both its long and short axes. To test the rotational motility of the kinesin-GNRs during translation, we performed the surface-gliding assay for ZEN4-GNRs, in which ZEN4-GNR particles were sparsely attached on the glass surface (3–8 particles per 100 μm²) (Materials and Methods). We found that the short microtubules unidirectionally rotated on ZEN4-GNRs whilst only their ends were in contact with the GNRs (Fig. 6 and Supplementary Movie 6). In the geometry of observation images, we observed both counterclockwise rotation ($n = 4$ microtubules) and clockwise rotation ($n = 4$ microtubules). Due to the property of ZEN4 dimer to accumulate at microtubule plus ends[26], these short microtubules continuously rotated with the microtubule tip attached to the ZEN4-GNR. These observations also support a model in which

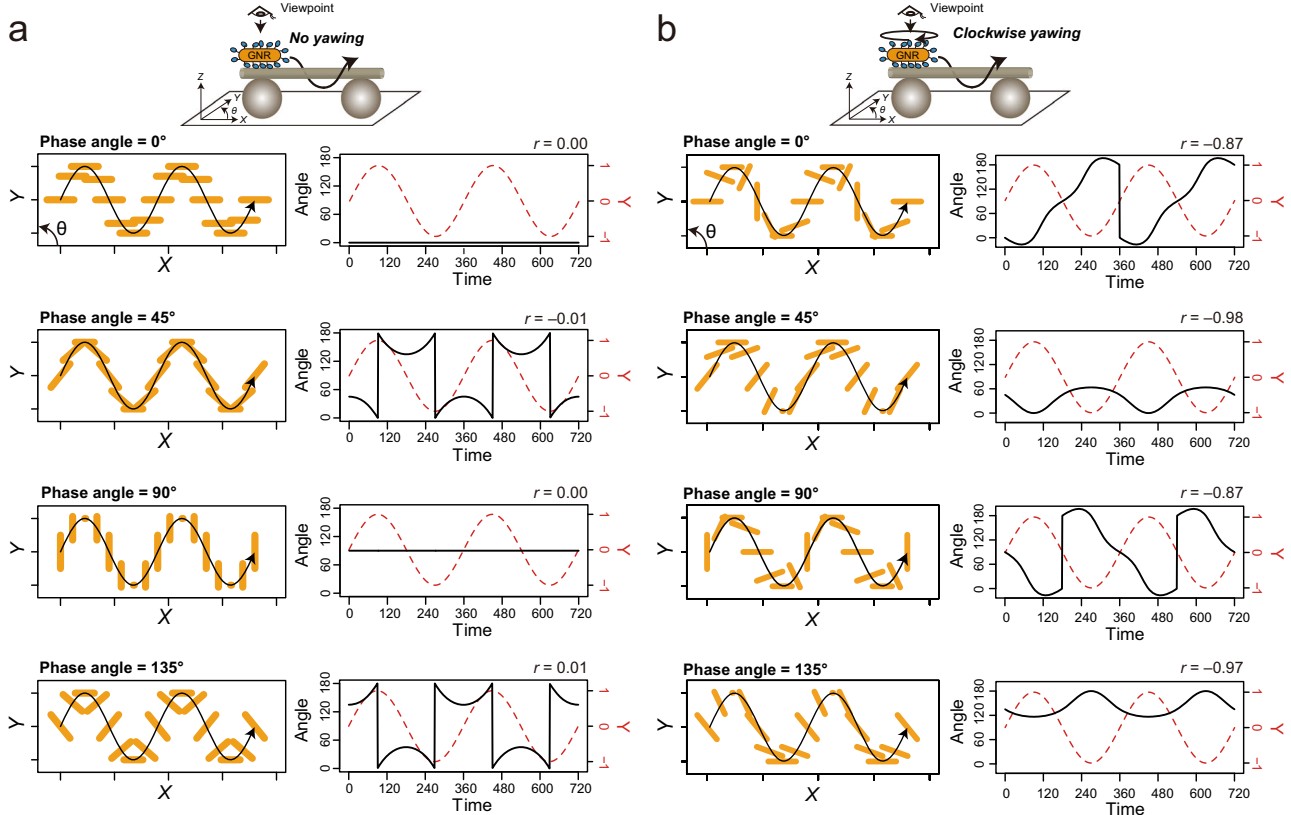

**Fig. 4 Interpretation of the observed polarization rotations and translational movement of the kinesin-coated gold nanorods (GNRs). a, b** Montage of the projection images of a GNR (orange bar) in the *X-Y* plane, obtained by a model of a motor-coated GNR with no yawing (**a**) or 180° clockwise yawing during one period of helical motion (**b**). Time trajectories of the *Y*-displacement (*Y* = −1 to 1) and angle (*θ* = 0 to 180°) of a motor-coated GNR with different initial phase angles (*φ* = 0°, 45°, 90°, and 135°). The value of *r* is Pearson's correlation coefficients between the *Y*-displacement and the angle. Trajectories of a motor-coated GNR in the case of counterclockwise yawing are shown in Supplementary Fig. 6. See also Supplementary Movies 2, 4, and 5.

kinesin-coated GNRs moving along a microtubule are driven to undergo both helical and yaw-axis rotations.

**Theoretical considerations of kinesin-GNR motility.** We found that the polarization and helix pitches were comparable among KIF1A- and ZEN4-GNRs (~0.7 μm) (*P* > 0.05 for Welch's *t* test), despite substantial differences in their forward velocities (*P* < 0.01 for adjusted pairwise comparisons using Welch's *t* test with the Holm method). The helix pitch is given by $P_{Helix} = 2\pi V_X/\omega_{Helix}$, where $P_{Helix}$ is the pitch size of helix, $V_X$ is the forward velocity, and $\omega_{Helix}$ is the angular velocity in the *Y-Z* plane. We see that the forward velocities are different among KIF1A- and ZEN4-GNRs but the pitches are comparable, so that the ratios of $V_X$ to $\omega_{Helix}$ are thus almost equal. This means that the angular velocity of helical motion is directly proportional to the forward velocity for two kinesin constructs, suggesting that the off-axis component is intrinsic to each power stroke. This relationship can be well modeled by a 2D noise-driven ratchet mechanism, as proposed by Mitra et al. for single-headed KIF1A[19]. Based on this mechanism, we now consider how yawing of the kinesin-coated GNRs influences its motility. Since a kinesin-coated GNR rotated 180° about its yaw axis in one period of a helical trajectory with the helix pitch of about 700 nm, the GNR rotates 2° ≈ 180°/(700 nm/8 nm) with 8 nm of forward displacement, which is equivalent to the length of the tubulin dimer. The 2° rotation generates at most ±1 nm ≈ ±20~34 nm × tan(2°) of lateral displacements of the kinesin molecules bound on the GNR (40 nm in diameter and 68 nm in length) (Fig. 7a). Lateral displacement of 1 nm is

equivalent to 0.2 ≈ 1 nm/5.1 nm[27] when the length between the binding sites toward the off-axis is set to 1, which would be sufficient to influence the noise-driven ratchet mechanism (Fig. 7b). A recent discussion by Mitra et al. indicates that around 1 μm of a helix pitch is reproduced by the 2D Brownian ratchet model with a slight off-axis asymmetry factor relative to the on-axis asymmetry factor[19].

We evaluated the off-axis fluxes of three particles differing in their starting positions for diffusion (−*d*, 0, +*d*) in the noise-driven ratchet model with asymmetry factor (*α*) after 1/*γ* units of time[28] (Materials and Methods):

$$J_0 = \frac{\gamma}{4}\left\{\text{erfc}\left[\frac{\alpha}{2}\sqrt{\gamma}\right] - \text{erfc}\left[\frac{1-\alpha}{2}\sqrt{\gamma}\right]\right\} \quad (3)$$

$$J_{\pm d} = \frac{\gamma}{4}\left\{\text{erfc}\left[\frac{\alpha \mp d}{2}\sqrt{\gamma}\right] - \text{erfc}\left[\frac{1-\alpha \pm d}{2}\sqrt{\gamma}\right]\right\} \quad (4)$$

where $J_0$ represents the flux of particles starting diffusion at 0 and $J_{\pm d}$ represents the flux of particles starting diffusion at ±*d* when the length between the binding sites toward the off-axis is set to 1. In calculations with *α* = 0.53 and *γ* = 160 s$^{-1}$ (Materials and Methods), the absolute values of $J_{+d}$ and $J_{-d}$ increased with *d* along a sigmoidal curve and the net flux, $J_{-d} + J_0 + J_{+d}$ described the convex downward function of which the value reached a minimum at *d* = 0.5 (Fig. 7c, d). Importantly, the slight displacement (*d* < 0.2) is sufficient to cause biased flux because |$J_{-d}$| is much larger than |$J_{+d}$| (Fig. 7c, inset). These suggest that unidirectional rotation of a motor-coated cargo enhances the

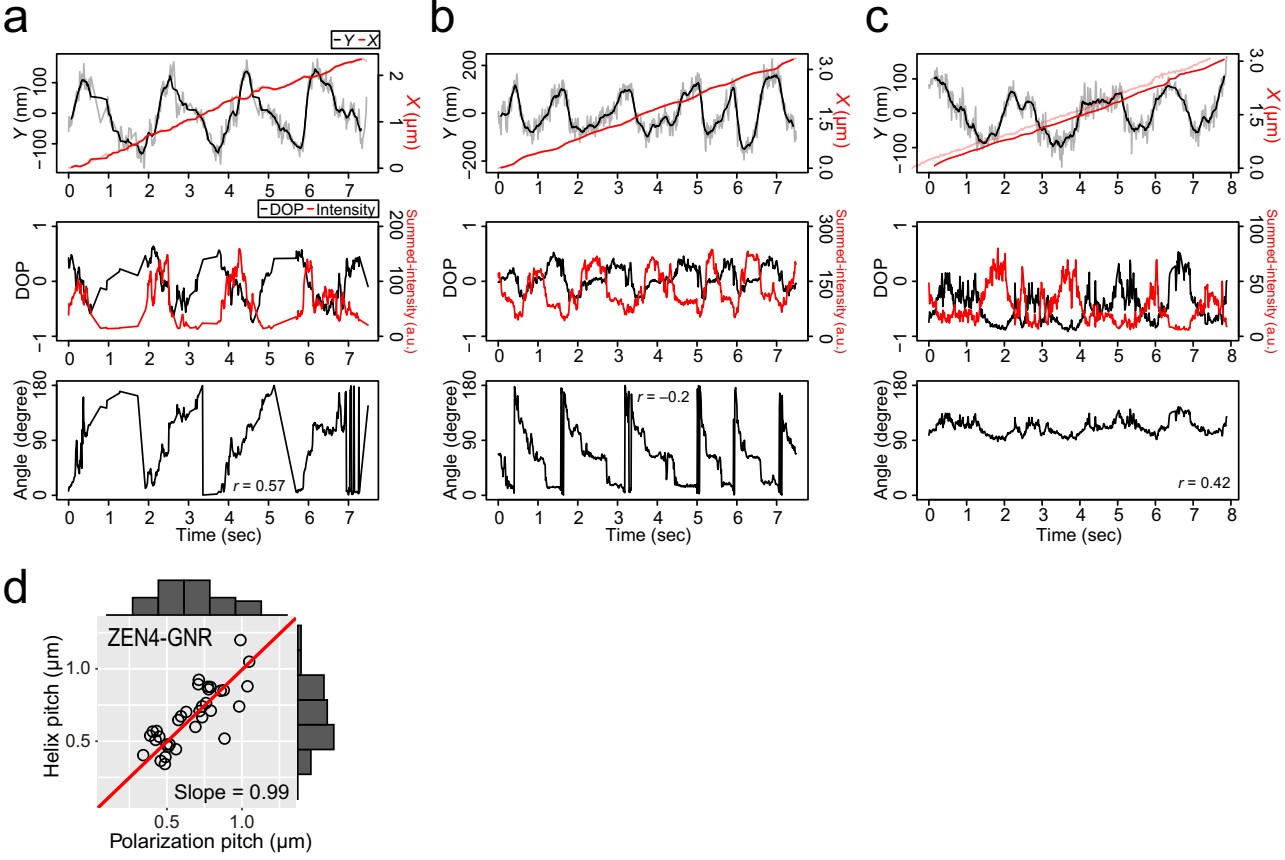

**Fig. 5 GNR motility assay for ZEN-4. a–c** Typical time trajectories of the *X* (red line)- and *Y* (black line)-displacements, the degree of polarization (DOP, black line) and summed-intensity (red line) of the two spots, and the estimated angles of ZEN-4-coated gold nanorods (ZEN4-GNRs). The three trajectories represent counterclockwise (**a**), clockwise (**b**), and oscillatory (**c**) rotations of the GNR polarization in the sample plane. The value of *r* is Pearson's correlation coefficient between the *Y*-displacement and the angle. **d** Distribution of the helix and polarization pitches of ZEN4-GNRs. The red lines represent linear fits ($y = ax$) of which the slopes (*a*) are 0.99 ($R^2 = 0.97$).

sideward bindings of motor molecules toward the biased direction.

To test this suggestion, we performed simple 2D Monte Carlo simulations. Multiple motor particles are linked to a rigid body cargo with a spring, stochastically taking biased forward and/or sideward steps on the 2D lattice, driving force and moment applied to the cargo, and generating additional torque ($T_{step}$) to rotate the cargo when taking a step (Fig. 7e). Displacement and rotation of the cargo were based on the force- and moment-balance equations (Materials and Methods). When $T_{step}$ was −24, the cargo exhibited periodical clockwise rotation (Fig. 7f). The rotation period of the cargo roughly matched the 13-lattice period of the *Y*-displacements modeled as the 13-protofilament microtubule (Supplementary Fig. 7). As the theory predicted, the lateral displacement of the cargo with torque generation increased compared to that without torque generation (Fig. 7g). In the experiments, as shown above, the pitches of the helical motion of KIF1A-GNR (~0.7 μm) and ZEN4-GNR (~0.7 μm) were shorter than those of the corkscrewing of microtubules on the surfaces of KIF1A (~0.9 μm) and ZEN-4 (~0.8 μm[11]), respectively. These results support the analysis by the noise-driven ratchet model showing that unidirectional rotation of a motor-coated cargo enhances the sideward bindings of motor molecules toward the biased direction.

**Discussion**

By tracking the intensities of polarized light reflected into orthogonal axes by motile kinesin-GNRs, we reveal periodic rotations of teams of cargo-attached low-processivity kinesins. For both KIF1A and ZEN-4 kinesins, around 20% of the kinesin-GNR runs showed periodic changes in the polarization signal, phase-locked to the orbit of the GNR around the microtubule, over multiple orbital periods. This periodic signal indicates periodic rotation of the GNR about its own short axis. We expect that only a fraction of runs will generate a detectable polarization signal, because for this to occur, the plane of yawing of the kinesin-GNRs about its short axis need to be nearly parallel to the longitudinal axis of the microtubule. Our data thus are consistent with all the GNR-kinesin particles rotating periodically around their short axes as they move along an overall helical trajectory toward the plus end of the microtubule.

How might this finding relate to kinesin-driven cargo transport in cells? Importantly, we expect cargo rotation only in the case that the kinesins are rigidly attached to their cargo. In cases where the cargo is a membrane-bounded vesicle or organelle, kinesin-adaptor complexes are thought to diffuse in the bilayer, so that any torque would dissipate. For other cargoes, for example, ribonuclear proteins, viruses, and IFT particles, kinesin binds rigidly to a protein component of the cargo, and torque is expected to transmit[29–32]. Specifically, centralspindlin, a stable tetramer formed by a dimer of a RhoGAP, CYK-4, and kinesin-6 (ZEN-4), moves along microtubules as clusters toward the midzone[26]. In these cases, we speculate that rolling and tumbling of cargoes may help the motor avoid obstacles.

Due to the relatively low aspect ratio of the GNRs we used, the elevation angles of the long axis of the kinesin-GNRs might tend

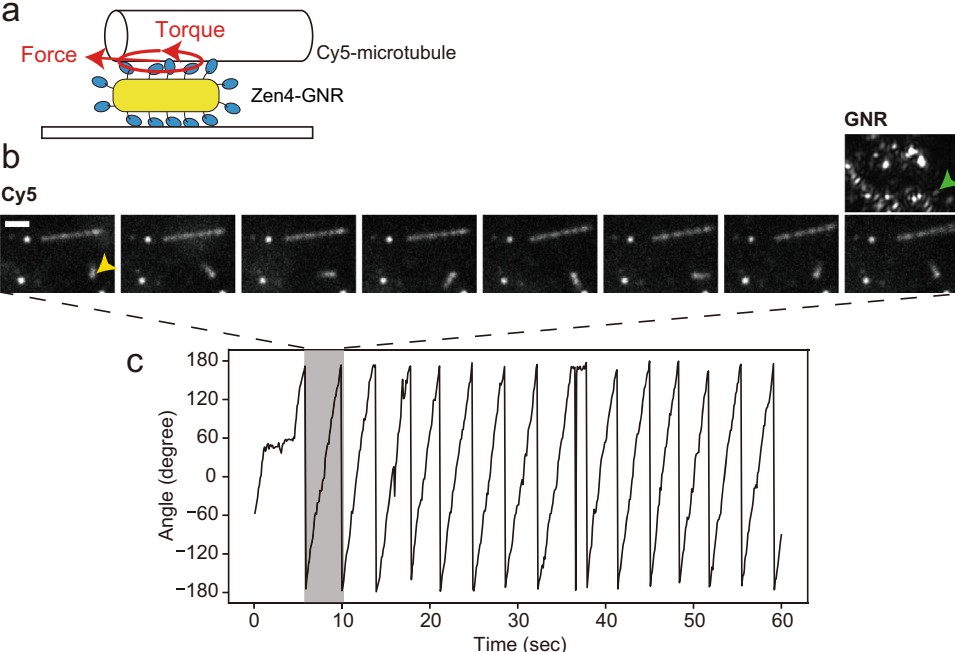

**Fig. 6 Surface-gliding assay for ZEN-4 coated gold nanorods (ZEN4-GNRs). a** Schematic drawing of the surface-gliding assay, whereby the generation of force and torque rotate a Cy5-labeled microtubule (Cy5-microtubule) on the ZEN4-GNR. **b** Montage of fluorescence images of Cy5-microtubules every six frames (600 ms), showing that the short Cy5-microtubule continuously rotated (yellow arrowhead). Scale bar 2 μm. GNR image of the same region-of-interest of the Cy5 images is also shown (green arrowhead). **c** Time trajectory of the microtubule orientation.

to change, resulting in unstable polarization fluctuations in the majority of the kinesin-GNRs. However, GNRs with higher aspect ratio may inhibit yawing of the kinesin-GNR, since alignment of the long axes of the microtubule and the kinesin-GNR will maximize the number of kinesin molecules interacting with the microtubule.

A recent single-molecule study of the kinesin-1 dimer construct revealed that it unidirectionally rotates the stalk domain by ~1° with 8-nm steps at high ATP conditions and in the presence of an external force[9]. Since the KIF1A-GNR and the ZEN4-GNR rotate 2° per 8-nm displacement ($\approx 180°/700$ nm × 8 nm), the torque generated by these kinesins might be comparable to that of kinesin-1. In solid mechanics, the twist angle of an object due to an applied torque is given by $\theta_{step} = T_{step}L/KG$, where $T_{step}$ is the torque, $L$ is the length of the object, $G$ is the modulus of rigidity of the object, and $K$ is the polar moment of inertia of the object[33]. The torque per step of kinesin-1 ($T_{step}$) was estimated to be 170 pN nm[9]. The polar moment of inertia of the motor molecule, approximated as a cylinder, is given by $K = \pi d^4/32$, where $d$ is the diameter of the cylinder. Because kinesin molecules were bound to the gold nanorod via a streptavidin, the motor molecule could be roughly approximated as a cylinder with a diameter ($d$) of 5 nm and length ($L$) of 12 nm. The modulus of rigidity of the motor molecule ($G$) is assumed to be 1 GPa at a Poisson ratio of 0.25, which is equivalent to the Young's modulus for proteins such as actin, tubulin, and coiled-coil (~2 GPa)[33]. We thus obtained 2° of the twist angle $\theta_{step}(= T_{step}L/KG)$, which is consistent with the rotation angle of kinesin-GNRs per 8-nm displacement. These data thus suggest that torque is generated by conformational changes in the kinesin motor domain. Previous structural studies of kinesin-1 and KIF1A revealed that rearrangement in the motor domain from the pre-stroke state to the post-stroke state generates global rotation of the motor domain[34–36]. The neck-to-tail region is critical for torque transmission from the head domain to the cargo. In the case of ZEN-4, its neck domain contains a non-canonical long sequence;

however, high-speed AFM imaging of a ZEN-4 dimer revealed a distinctive globular mass in the neck region[37]. Thus, the non-canonical neck-domain of ZEN-4 may be a stiffness component rather than a flexible linker.

In the GNR assay experiments, each kinesin-coated GNR exhibited specific directionality of yawing, which is consistent with previous single-molecule measurements for the kinesin-1 dimer[9]. Although the detailed mechanism of directional yawing of a kinesin team remains unclear, not only the power stroke in the motor domain but also the formation of kinesin molecules on the GNR and further the configuration of the kinesin-GNR and the microtubule might be involved.

We envision that the GNR rolls as it glides over the microtubule surface, driven by the same forces that cause microtubules to roll as they slide on coverslips coated in non-processive kinesins. This rolling has no impact on the instantaneous forces because we assume a uniform density of motors on the GNR. To achieve this, we sought to saturate kinesin binding to the GNR. Indeed inhomogeneous distributions of motors around the GNR might impact the trajectory of the GNR, because the amount of torque would vary. In all situations where a team of weakly processive kinesins is engaged, torque will be produced in proportion to the number of engaged motors.

Our in vitro GNR assay of monomeric KIF1A and dimeric ZEN-4 revealed that in kinesin teams, the production of torque drives not only orbital motion of the kinesin team around the microtubule axis, but also coupled yaw-axis rotation of the team and its attached GNR cargo. The coupled yawing and rolling rotations may be a special case of kinesin motility that could be observed in in vitro GNR assay. More importantly, however, our experiments and theoretical analyses implied that yawing of rigidly-bound kinesin cargo facilitates lateral movement. Previous studies proposed that sideward stepping of kinesins and dynein allows them to bypass roadblocks on a microtubule[6–8,11,13,38]. When a cargo hits a roadblock and bypasses it on a microtubule, the translational motion would be temporarily slowed but the

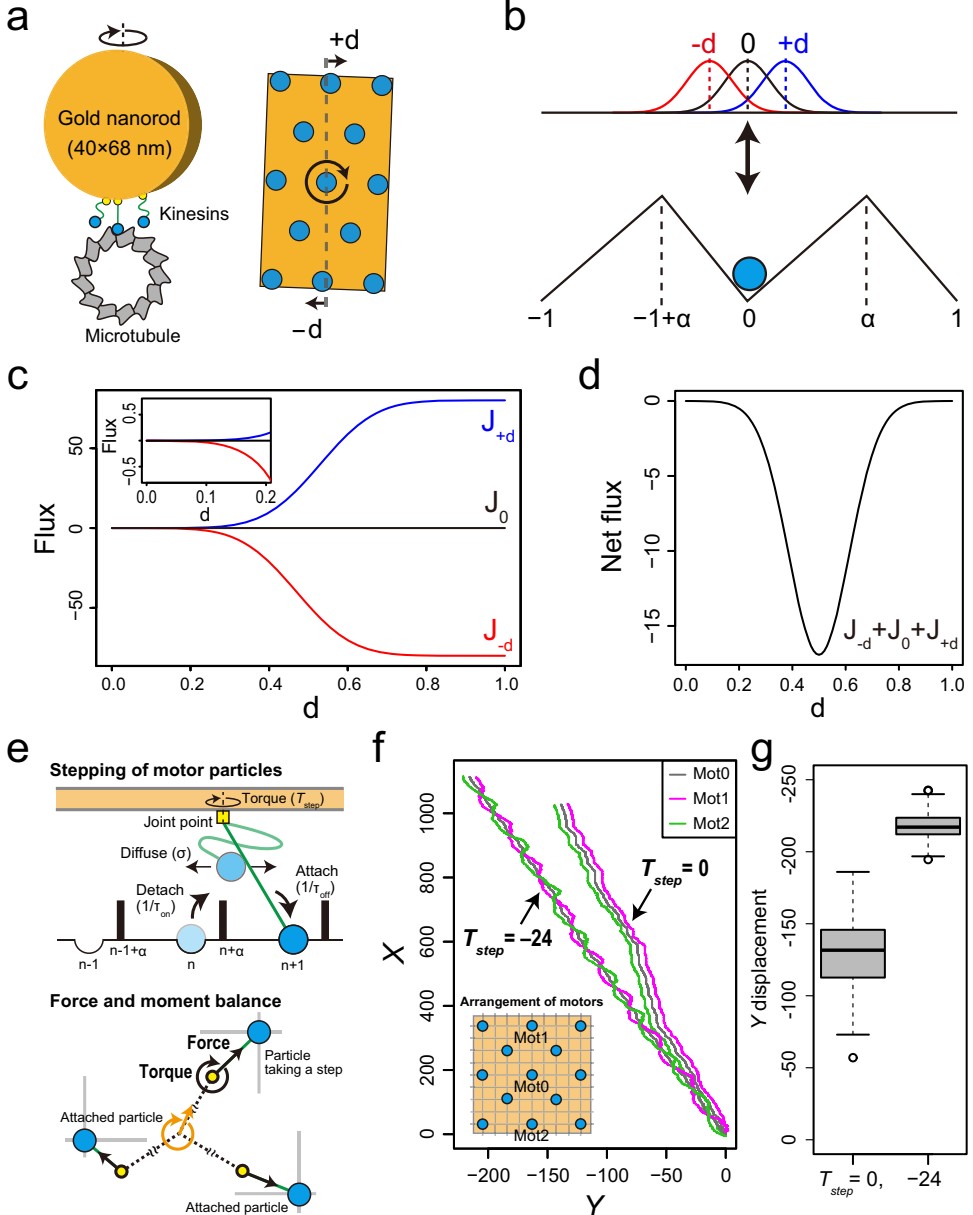

**Fig. 7 Mechanistic model for the yawing and helical motion of kinesin-coated gold nanorods (GNRs). a** By rotation of the GNR, motors in the posterior and anterior positions relative to the rotation axis move laterally with displacement of $\pm d$. **b** This situation can be modeled using the noise-driven ratchet model with asymmetry potentials ($\alpha$) and three different starting positions of the particles ($-d$, 0, $+d$). **c, d** Fluxes of three particles at the starting positions of diffusion in the noise-driven ratchet with asymmetry potentials after $1/\gamma$ units of time (Materials and Methods). **c** Fluxes of three particles, $J_{-d}$, $J_0$, and $J_{+d}$ as a function of d with $\alpha = 0.53$ and $\gamma = 160$ s$^{-1}$. **d** Net flux, $J_{-d} + J_0 + J_{+d}$ as a function of d. **e–g** 2D Monte Carlo simulation of 13 motor particles without and with torque generation ($T_{step} = 0$ and $-24$, respectively). **e** Schematic description of the model. **f** Typical $X$-$Y$-trajectories of the forward, and center, and backward particles (magenta, gray, and green lines, respectively) represent biased forward, sideward and rotational motions of the cargo. **g** Box plot of comparison of lateral displacements without and with torque generation. The box represents the 75-25th percentiles, and the median is indicated as black line ($n = 300$ simulations). The circle sign represents an outlier.

rotational motion be allowed. The GNR imaging of endosomes in cells showed that the rotational states of endosomes correlate with changes in translational speed[39]. Therefore, the translational and rotational motilities of motor proteins might complement each other for their transport functions.

Taken together, our detection of biaxial rotation of cargoes transported by teams of rigidly-attached weakly processive kinesins reveals a new, to our knowledge, aspect of kinesin-driven transport[36,40], and suggests in particular that rolling and tumbling of cargo might be relevant to intracellular function[39].

## Materials and methods

**Gold nanorod imaging.** Figure 1a shows a schematic drawing of the laser dark-field microscope system used for gold nanorod (GNR) imaging. The laser dark-field optical system was constructed on an inverted microscope (Ti; Nikon, Tokyo, Japan), equipped with variable-angle epi-illumination[21,22] and a custom-built perforated dichroic mirror (PDM) (Sigma Koki, Tokyo, Japan). The PDM had an elliptical hole (short axis = 6 mm; long axis = 8.5 mm) at its center position, which made a circular window when viewed from the optical axis. The scattered light from the GNRs was transmitted through this central circular window. Illumination was performed using a 635-nm diode laser (MRL-III-635L-10mW; CNI laser, Changchun, China), which was further attenuated by neutral density filters (ND) and converted to circularly polarized light by a quarter-wave plate (WPQ-6328-4M; Sigma Koki, Tokyo, Japan).

The collimated and circularly polarized incident beam was focused by a lens (L1 in Fig. 1a, $f = 220$ mm) onto the back focal plane of the objective lens (PlanApo N, 60×, NA 1.42; Olympus, Tokyo, Japan). GNR images were obtained by laser dark-field imaging using highly inclined illumination with nearly total internal reflection angle. The back focal plane was focused outside a camera port of the microscope with an achromatic lens (L3 in Fig. 1a, $f = 60$ mm) to make an equivalent back focal plane for constructing the optics for polarization measurements and 3D tracking.

For polarization measurements, the scattered light from a GNR illuminated by polarized incident beam was further split into two orthogonally polarized components by a polarization beam displacer (BD40; Thorlabs, Tokyo, Japan), which was set in the optical path between the relay optics constructed by two lenses (L3 ($f = 60$ mm) and L4 ($f = 120$ mm) in Fig. 1a)[41]. Two separated and polarized light beams were imaged as two spots on a CMOS camera (LDH2500; Digimo, Tokyo, Japan). The orientations of the two orthogonally separated polarization components of the beam displacer were diagonal relative to the transmission axis of the polarization filter. For angular calibration, a linear polarizer was additionally set at the position between L2 and L3 and rotated from 0° to 180° in 10° intervals (Fig. 1b).

For 3D tracking, concave and convex cylindrical lenses (CnC ($f = -200$ mm) and CvC ($f = 200$ nm)) were set in front of the camera[24]. The interval between CnC and CvC was 13 mm. Calibration of 3D tracking along the Z-axis was performed by moving the objective in 50-nm intervals (Supplementary Fig. 1), using a custom-built stage equipped with a pulse motor (SGSP-13ACT; Sigma Koki, Tokyo, Japan) and a controller (QT-CM2; Chuo Precision Industrial, Tokyo, Japan). We analyzed the widths of the GNR spots along the X-axis ($S_X$) and the Y-axis ($S_Y$) obtained by 2D-Gaussian fits. The calibration factor for the Z-displacement obtained from the ratio of $S_Y$ to $S_X$ was 0.98 μm/ratio.

The temperature of the sample stage was controlled using a custom-built water-circulating temperature control system. Image acquisition in the GNR motility assay was performed at 100 frames/s with the commercial software for the CMOS camera (Digimo, Tokyo, Japan). The data were saved as eight-bit AVI files. The effective pixel size of the detector was 42 nm. Image analysis and further analyses were performed using custom software written in LabVIEW (National instruments, Tokyo, Japan).

**Polarization measurements.** For polarization measurements, the scattered light from a GNR illuminated by polarized incident beam was split into two polarized components by the polarization beam displacer as mentioned above. Thus, the intensities of two polarized GNR spots as functions of the GNR orientation ($θ$) are expressed by

$$I_{bottom} = I_0 \cdot \cos^2(θ + ζ) \cdot \cos^2(θ + ζ - η) + b \quad (5)$$

$$I_{upper} = I_0 \cdot \cos^2(θ + ζ) \cdot \cos^2(θ + ζ - η + π/2) + b \quad (6)$$

where $I_0$ is the amplitude; $θ$ is the GNR orientation in the sample plane; $ζ$ and $η$ are phase angles of polarization of the incident beam in the sample plane and the beam displacer, respectively; and $b$ is the background intensity. The observed intensity profiles in Fig. 1c were well fitted by these functions with $I_0 = 127.5 \pm 3.3$, $ζ = 99.3° \pm 0.8°$, $η = 45.3° \pm 0.7°$, and $b = 19.5 \pm 0.9$. GNR angles in the sample plane were obtained by fitting an ellipse function to the plot of the degree of polarization $(I_{bottom} - I_{upper})/(I_{bottom} + I_{upper})$ as a function of the summed-intensity $(I_{bottom} + I_{upper})$, as shown in Fig. 1d.

**Kinesin constructs.** In all kinesin constructs used in this study, the functional sequence Avi-His, which consists of the Avi-tag (GLNDIFEAQKIEWHE) for biotinylation and the 6×His-tag for histidine-affinity purification, was fused to the C-terminus of truncated kinesin domains. For the monomeric KIF1A construct, the fragment of human KIF1A (1–366 amino acid residues (AAs)) obtained from a human cDNA library was cloned into eGFP-Avi-His/pColdIII plasmids. eGFP stabilized the solubility of the truncated KIF1A sample. In addition, for the surface-gliding assay of the KIF1A construct, we used the construct of human KIF1A (1–366 AAs) cloned into mRuby-Zdk1-His/pColdIII plasmids. For the dimeric ZEN-4 construct, the fragment of C. elegans ZEN-4 (1–555 AAs) obtained from CBD-TEV-ZEN4/pGEX-6p plasmids[37] was cloned into Avi-His/pET32a plasmids. For the dimeric kinesin-1 constructs, the fragments of rat kinesin-1 (1–430 AAs for the dimeric constructs[42]) were cloned into Avi-His/pColdIII plasmids. All constructs were verified by DNA sequencing.

**Expression and purification of kinesin.** For expression and purification of the kinesin constructs, the proteins were expressed in Escherichia coli strain BL21 Star (DE3) cells using 0.4 mM Isopropyl-β-D-thiogalactopyranoside (IPTG) (Sigma) for 9 to 12 h at 20 °C. The collected cells were lysed in lysis buffer (50 mM potassium phosphate; pH 7.4, 350 mM NaCl, 1 mM MgCl₂, 10 μM ATP, 1 mM DTT, 0.1% Tween20, 10% glycerol, protease inhibitors, 50 mM imidazole) and sonicated in iced water for 15 min. The bacterial lysate was centrifuged for 20 min at 260,000 g at 4 °C to remove cell debris and insoluble proteins. The clarified supernatant was purified with the His-tag affinity by chromatography with a HisTrap HP Ni²⁺sepharose column (GE Healthcare). The collected peak fraction was further purified through a HiTrap desalting column (GE Healthcare) to exchange the buffer

(20 mM potassium phosphate; pH 7.4, 1 mM MgCl₂, 1 mM DTT, 10 μM ATP) containing 80 mM NaCl for the KIF1A and the kinesin-1 constructs, or 250 mM NaCl for the ZEN-4 construct.

For the KIF1A construct, we further purified the active proteins based on their affinity for microtubules. KIF1A and microtubules were mixed and incubated for 15 min in the presence of 1 mM AMPPNP. The mixture was centrifuged for 20 min at 260,000 g at 25 °C to remove unbound KIF1A molecules. The pellet was resuspended in ATP-containing buffer (20 mM PIPES-KOH; pH 7.4, 5 mM MgCl₂, 10 mM potassium acetate, 250 mM NaCl, 20 μM paclitaxel, 5 mM ATP) and incubated for 15 min at 23 °C for detachment of KIF1A molecules from the microtubules. Microtubules were removed by centrifugation (260,000 g, 15 min, 23 °C).

The purified proteins were flash-frozen and stored in liquid nitrogen. The concentration of proteins was estimated using the Bradford method with the use of BSA as a standard.

**Purification of tubulin.** Tubulin was purified from porcine brains through four cycles of temperature-regulated polymerization and depolymerization in a high-molarity PIPES buffer to remove contaminating MAPs[43]. The purified tubulin was flash-frozen and stored in liquid nitrogen.

**Preparation of Cy5-microtubules.** Tubulin stock solution (~6 mg mL⁻¹, 50 μL) and Cy5-labeled tubulin solution (~20 mg mL⁻¹, 1 μL, ~50% labeled) were mixed on ice and then diluted twofold with chilled BRB80 buffer (80 mM PIPES-KOH; pH 6.8, 1 mM MgCl₂, 1 mM EGTA). One microliter each of GTP (100 mM), and MgCl₂ (100 mM) was added to the diluted tubulin solution. The tubulin solution was incubated at 37 °C for 1 h for polymerization and stabilized with 20 μM paclitaxel. The Cy5-microtubules were further purified by centrifugation. Ray et al. reported that polymerization in PIPES buffer results in ~60% of 14-protofilament microtubules and ~30% of 13-protofilament microtubules[44].

**Preparation of protein-G-coated beads and streptavidin-coated gold nanorods.** For preparation of protein G-coated beads, carboxylate-modified fluorescent microbeads (0.5 μm; Thermo Fisher Scientific, USA) were diluted in the activation buffer (100 mM MES; pH 6.0). EDC and sulfo-NHS were added to the bead solution and incubated for 15 min at room temperature. The beads were then washed with 100 mM HEPES buffer at pH 8.0. Protein-G was then added to the bead solution and reacted for 2 h at room temperature. Any unreacted amine reactive groups were quenched by adding excess glycine. Excess proteins were removed by centrifugation (21,000 g, 15 min, 23 °C). The beads were resuspended in BRB80 buffer.

For preparation of streptavidin-coated GNRs, biotin-labeled GNRs (5 nM) (C12-40-600-TB; Nanopartz, CO, USA) and streptavidin (5 mg mL⁻¹) were mixed. After 15 min incubation, excess streptavidin was removed by centrifugation (4000 g, 3 min, 23 °C) and then the pellet was resuspended in Milli-Q water and sonicated for 2 s. The procedure for removing excess streptavidin was repeated twice.

**GNR motility assays.** Streptavidin-coated GNR (2 μL, 5 nM) and kinesin sample (2 μL, 1–2 μM) were mixed and incubated for at least 15 min at room temperature. For the assay of KIF1A, 2 μL of 0.2% Tween20 was further added to the mixture of kinesin and GNR. For the assay for ZEN-4, 0.5 μL of 5 M NaCl and 2 μL of 0.2% Tween20 were further added to the mixture of kinesin and GNR.

A flow chamber (sample volume of about 5 μL) was prepared using two coverslips (24 × 32 mm and 18 × 18 mm) stuck by double-sided tape. Protein-G-coated bead solution was nonspecifically absorbed to the coverslip. After 3 min incubation, the flow chamber was washed with 20 μL of 1 mg mL⁻¹ BSA and further incubated for 3 min. The flow chamber was washed with 20 μL of BRB80. Then, 10 μL of 20 μg mL⁻¹ anti-β -tubulin antibody solution (sc-58884; Santa Cruz, USA) was flowed to the flow chamber. After 3 min incubation, the flow chamber was washed with 20 μL of BRB80. Then, 10 μL of Cy5-labeled microtubule solution (~2 μM tubulin concentration) was flowed into the flow chamber such that the microtubules bound to the beads (Fig. 1e). After 3 min incubation, the flow chamber was washed with 20 μL of 1 mg mL⁻¹ casein. After 3 min incubation, the flow chamber was washed with 20 μL of the motility buffer (20 mM PIPES-KOH; pH 7.4, 4 mM MgCl₂, and 10 mM potassium acetate) containing the kinesin-coated GNRs (~100 pM), ATP (2 mM), the ATP regeneration system (0.8 mg mL⁻¹ creatine kinase, 10 mM creatine phosphate), the oxygen scavenger system (3 mg mL⁻¹ glucose, 50 U mL⁻¹ glucose oxidase, 1200 U mL⁻¹ catalase), and 1 mg mL⁻¹ casein. The GNR assays were performed at 27 °C. Image acquisition was performed at 100 frames s⁻¹.

**Surface-gliding assays for KIF1A with QD-coated microtubules.** Biotinylated (biotin-(AC₅)₂ Sulfo-OSu, Dojindo, Kumamoto, Japan), Cy5-labeled microtubules were prepared by co-polymerizing biotinylated, Cy5-labeled and non-fluorescent tubulin in a molar ratio of 1:3:75 in BRB80 with 1 mM GTP and 1 mM MgCl₂ for 30 min at 37 °C and then stabilized by 20 μM taxol. QD-coated microtubules were prepared by adding 13 nM streptavidin-coated QD (Qdot 525 streptavidin conjugate, Thermo Fisher Scientific) diluted in BRB80 containing 20 μM taxol

(BRB80T) and incubated for 30 min with an equal volume of 4 mg mL$^{-1}$ micro-tubules and then diluted to 33 pM QD and 10 µg mL$^{-1}$ microtubules in BRB80T containing 0.4 mg mL$^{-1}$ α-casein (Sigma-Aldrich)[18]. The surface-gliding assays were performed in flow chambers assembled from two coverslips attached using double-sided tapes. Before the assay, kinesin stock solution was diluted to 1 µM with motility buffer. Five microliters of a 5 mg mL$^{-1}$ solution of protein G (Sigma-Aldrich) was added to the flow chamber (about 5 µL). After 5 min incubation, the flow chamber was washed with 20 µL of BRB80 buffer, followed by another 5 min of incubation with 5 µL of 0.05 mg mL$^{-1}$ anti-His-tag antibody. The flow chamber was washed with 20 µL of BRB80 buffer, and then 5 µL of 0.5 mg mL$^{-1}$ casein solution was added. After 5 min incubation, the flow chamber was washed with 20 µL of BRB80 buffer. Then, 10 µL of KIF1A solution (1 µM) was added to the flow chamber. The flow chamber was then washed with 20 µL of BRB80 after 5 min incubation, and 20 µL of QD-coated microtubule solution (0.05–0.1 µM tubulin) was added to the flow chamber. Finally, the flow chamber was washed with 20 µL of BRB80 buffer after 5 min incubation, and 20 µL of motility buffer containing 3 mM ATP, 20 µM paclitaxel, 0.5% β-mercaptoethanol, 0.25 mg mL$^{-1}$ casein, ATP regeneration and oxygen scavenger systems were added. Fluorescent images of the QD- and Cy5-labeled microtubules were captured using TIRF microscopy at 25 °C. Corkscrewing pitches in the surface-gliding assay were determined by measuring the longitudinal displacement of the QDs in each period of oscillation of the lateral displacement along the microtubule axis.

**Surface-gliding assays for ZEN4-GNR.** Preparations of Cy5-labeled microtubules and ZEN4-GNR were prepared as described above. Surface-gliding assays were performed in flow chambers assembled from two coverslips attached using double-sided tape. The ZEN4-GNR solution was diluted to ~50 pM with motility buffer and added to the flow chamber (about 5 µL). After 5 min incubation, the flow chamber was washed with 20 µL of BRB80 buffer, followed by 5 min incubation with 10 µL of 0.5 mg mL$^{-1}$ casein solution. The flow chamber was then washed with 20 µL of BRB80 buffer, and 20 µL of Cy5-labeled microtubule solution (2 mM ATP, Cy5-labeled microtubules, 0.25 mg mL$^{-1}$ casein, and ATP regeneration and oxygen scavenger systems in motility buffer) were added. The assays were performed at 27 °C. Image acquisition of the Cy5-labeled microtubules was performed at 10 frames s$^{-1}$.

**Quantification and statistical analysis.** Polarization and position data of the observed kinesin-coated GNRs were obtained by image analysis with software written in LabVIEW (National instruments, Tokyo, Japan). Velocities of the kinesin-coated GNRs were determined by fitting the trace of the longitudinal displacement along the microtubule axis (X-direction) with a linear function. Rotation and helix pitches in the GNR assays were determined by measuring the X-displacements in the periods of oscillations of the angle- and Y-trajectories, respectively. Statistical hypothesis testing was performed using R statistical software.

**Correction of the measured pitch.** The measured pitch ($P_{\text{Measured}}$) comprised two helical elements: the helix pitch exerted by motors ($P_{\text{Motor}}$) without supertwist of the microtubule, and the supertwist of the microtubule depending on the number of the protofilaments ($P_{\text{MT}}$). $P_{\text{Measured}}$ is given by

$$\frac{1}{P_{\text{Measured}}} = \frac{1}{P_{\text{Motor}}} + \frac{1}{P_{\text{MT}}} \tag{7}$$

The PMT of 12-, 13-, and 14-protofilament microtubules is estimated to be about −3.4, −24.8, and 6.8 µm, respectively (we assign a positive value to the pitch in the left-handed helix)[44]. When the absolute value of $P_{\text{Measured}}$ is much smaller than the absolute value of $P_{\text{MT}}$ ($< \sim 1$ µm), $P_{\text{Motor}}$ is not appreciably changed by the supertwist (Supplementary Fig. 4).

**Mathematical model of biaxial rotations of a kinesin-coated GNR on a microtubule.** The GNR motility assay revealed that KIF1A- and ZEN4-GNRs unidirectionally rotate about the GNR's short axis, moving along the left-handed helix track. The helix track of the center position of a GNR is given by

$$helix\big(t,\, v,\, \omega_{\text{Helix}}\big) = \begin{pmatrix} X(t) \\ Y(t) \\ Z(t) \end{pmatrix} = \begin{pmatrix} vt \\ h \cdot \cos(\pi/2 - \omega_{\text{Helix}}t) \\ h \cdot \sin(\pi/2 - \omega_{\text{Helix}}t) \end{pmatrix} \tag{8}$$

where $v$ is the translational velocity along the X-axis, $\omega_{\text{Helix}}$ is the angular velocity in the Y-Z plane, and $h$ is the radius of the helix. Assuming that the long axis of the GNR is parallel to the X-Y plane under the initial conditions, the initial position matrix $p_0$ consisting of the tip, center, and end positions of the GNR is given by

$$p_0(\text{tip, center, end}) = \begin{pmatrix} r \cdot \cos(\varphi) & 0 & r \cdot \cos(\varphi + \pi) \\ r \cdot \sin(\varphi) & 0 & r \cdot \sin(\varphi + \pi) \\ 0 & 0 & 0 \end{pmatrix} \tag{9}$$

where $r$ is the radius along the long axis of the GNR and $\varphi$ is its phase angle. The experimental results suggest that a kinesin-coated GNR unidirectionally rotates about its short axis and further around the microtubule, so that the rotation

comprises yawing and rolling. The elemental rotation matrices are thus given by

$$R_Z(\omega_{\text{Yaw}}t) = \begin{pmatrix} \cos(\omega_{\text{Yaw}}t) & -\sin(\omega_{\text{Yaw}}t) & 0 \\ \sin(\omega_{\text{Yaw}}t) & \cos(\omega_{\text{Yaw}}t) & 0 \\ 0 & 0 & 1 \end{pmatrix} \tag{10}$$

$$R_Y(0) = \begin{pmatrix} 1 & 0 & 0 \\ 0 & 1 & 0 \\ 0 & 0 & 1 \end{pmatrix} \tag{11}$$

$$R_X(\omega_{\text{Roll}}t) = \begin{pmatrix} 1 & 0 & 0 \\ 0 & \cos(\omega_{\text{Roll}}t) & -\sin(\omega_{\text{Roll}}t) \\ 0 & \sin(\omega_{\text{Roll}}t) & \cos(\omega_{\text{Roll}}t) \end{pmatrix} \tag{12}$$

where $\omega_{\text{Yaw}}$ and $\omega_{\text{Roll}}$ are the angular velocities of yawing and rolling, respectively. The rotation matrix of the extrinsic element rotation sequence for our proposed motion is given by $R_X R_Y R_Z$. The helical and rotational trajectory of a motor-coated GNR is thus given as Eq. (1). By calculating Eq. (1),

$$\text{tip}(t) = \begin{pmatrix} X_{\text{tip}}(t) \\ Y_{\text{tip}}(t) \\ Z_{\text{tip}}(t) \end{pmatrix} = \begin{pmatrix} r \cdot \cos(\omega_{\text{Yaw}}t + \varphi) + vt \\ r \cdot \cos(\omega_{\text{Roll}}t) \cdot \sin(\omega_{\text{Yaw}}t + \varphi) + h \cdot \cos(\pi/2 - \omega_{\text{Helix}}t) \\ r \cdot \sin(\omega_{\text{Roll}}t) \cdot \sin(\omega_{\text{Yaw}}t + \varphi) + h \cdot \sin(\pi/2 - \omega_{\text{Helix}}t) \end{pmatrix} \tag{13}$$

$$\text{center}(t) = \begin{pmatrix} X_{\text{center}}(t) \\ Y_{\text{center}}(t) \\ Z_{\text{center}}(t) \end{pmatrix} = \begin{pmatrix} vt \\ h \cdot \cos(\pi/2 - \omega_{\text{Helix}}t) \\ h \cdot \sin(\pi/2 - \omega_{\text{Helix}}t) \end{pmatrix} \tag{14}$$

$$\text{end}(t) = \begin{pmatrix} X_{\text{end}}(t) \\ Y_{\text{end}}(t) \\ Z_{\text{end}}(t) \end{pmatrix} = \begin{pmatrix} -r \cdot \cos(\omega_{\text{Yaw}}t + \varphi) + vt \\ -r \cdot \cos(\omega_{\text{Roll}}t) \cdot \sin(\omega_{\text{Yaw}}t + \varphi) + h \cdot \cos(\pi/2 - \omega_{\text{Helix}}t) \\ -r \cdot \sin(\omega_{\text{Roll}}t) \cdot \sin(\omega_{\text{Yaw}}t + \varphi) + h \cdot \sin(\pi/2 - \omega_{\text{Helix}}t) \end{pmatrix} \tag{15}$$

Therefore, the time trajectory of the GNR orientation in the X-Y plane is given by

$$\frac{Y_{\text{tip}}(t) - Y_{\text{end}}(t)}{X_{\text{tip}}(t) - X_{\text{end}}(t)} \tag{16}$$

and thus Eq. (2) is obtained. To synchronize the rolling and the helix track, the angular velocity of the rolling ($\omega_{\text{Roll}}$) is identical to that of the helix track ($\omega_{\text{Helix}}$). And the angular velocity of the yawing ($\omega_{\text{Yaw}}$) is half that of the helix track because the GNR rotates half during one period of the helix track. The calculated helical and rotational trajectories of a kinesin-coated GNR and the time trajectories of the GNR orientation with different phase angles are shown in Fig. 4 and Supplementary Fig. 6, and Supplementary Movies 2, 4, and 5.

**Evaluation of particle flux in the noise-driven ratchet model.** The previous study by Astumian and Bier[28] clarified the particle flux in the two-state, noise-driven ratchet model. Following their discussions, we evaluated the particle flux for different start positions, because sideward translations (±d) of motors takes place owing to the rotation of their cargo (see Fig. 7a). The parameter $d$ is normalized as the length between the binding sites toward the off-axis is set to 1. In the two-state ratchet model, when the barrier moves to the down state, the particles start to diffuse. The probability density function of a diffusing particle at the start position of +d in the X-direction is given by

$$P_{+d}(x; t) = \frac{1}{2\sqrt{\pi t}} \exp\left[-\frac{(x-d)^2}{4t}\right] \tag{17}$$

The motion is driven by a thermal force, $F$, whose value is $-\gamma v$ where $\gamma$ is the drag coefficient and $v$ the velocity. The time constant at which the velocity approaches the terminal velocity (F/$\gamma$) is $1/\gamma$ (we assume that the diffusing particle has mass 1). After $1/\gamma$ units of time, the probability that the particle is located to the right of $\alpha$ is given by

$$\int_{\alpha}^{\infty} P_{+d}(x; 1/\gamma)dx = \frac{\gamma}{4}\text{erfc}\left[\frac{\alpha - d}{2}\sqrt{\gamma}\right] \tag{18}$$

and the probability that it on the left of $-1+\alpha$ is given by

$$\int_{-\infty}^{-1+\alpha} P_{+d}(x; 1/\gamma)dx = \frac{\gamma}{4}\text{erfc}\left[\frac{1 - \alpha + d}{2}\sqrt{\gamma}\right] \tag{19}$$

The particle flux, $J_{+d}$ (Eq. (4)) is given by Eq. (18) and Eq. (19). Similarly, $J_{-d}$ (Eq. (4)) and $J_0$ (Eq. (3)) are also obtained.

Following the probability density function of a diffusing particle, the standard deviation of diffusion is given by $\sqrt{2/\gamma}$. When $\gamma$ is 160 s$^{-1}$, $\sqrt{2/\gamma} \approx 0.11$, which is equal to the value of $\sigma$ in the Monte Carlo simulations (see below).

**Monte Carlo simulation**. We performed Monte Carlo simulations to model the observed rotational and helical motion. In the simulation, each particle is linked to a rigid body as a cargo with a spring (spring constant, $k$) and the joint points linking the particles and the cargo are fixed. The particles stochastically attach and dissociate from the rail (on-time constant, $\tau_{on}$; off-time constant, $\tau_{off}$). A detached particle diffuses around the joint point (standard deviation of diffusion, $\sigma$) in the X-Y-plane. When the diffusing particle overrides the asymmetric barrier (asymmetric factors, $\alpha_X$ and $\alpha_Y$) at the last time of diffusion, the particle takes a step in the X-Y-direction. The force- and moment-balance equations are given by $\sum_i^{N_{on}} k \vec{d_i} = 0$ and $\sum_i^{N_{on}} k\vec{d_i} \times \vec{l_i} - T_{step} N_{step} = 0$, where $k$ is the spring constant; $\vec{d_i}$ is an extension vector of the spring of the $i$-th particle that is bound to the rail; $\vec{l_i}$ is the position vector of the $i$-th joint relative to the center of the joint points connecting the particles that are bound to the rail; and $N_{on}$ is the number of the particles bound to the rail; $T_{step}$ is the torque per step of single motor particle; $N_{step}$ is the number of motor particles taking a step in the unit time.

The simulation was developed in LabVIEW (National Instruments). Pseudo code of the simulation can be found in Supplementary Note 1. The values for the parameters used in the simulation were $k = 1$; $\tau_{on} = 40$ units of time; $\tau_{off} = 10$ units of time; $\alpha_X = 0.12$ units of lattice size; $\alpha_Y = 0.58$ units of lattice size; $\sigma = 0.11$ units of lattice size; $T_{step} = 0$ or $-24$. The parameters for the biased forward and leftward displacements ($\tau_{on}$, $\tau_{off}$, $\alpha_X$, $\alpha_Y$, $\sigma$) were optimized when $T_{step}$ was 0 so that the ratio of forward to leftward steps is approximately 100:13 that is equivalent to the 0.8-μm helix pitch of KIF1A and ZEN-4 in the gliding assay (Fig. 7f). The value of $\sigma$ was determined by the evaluation of particle flux in the noise-driven ratchet model mentioned in the movie subsection.

**Statistics and reproducibility**. Statistical analysis of two groups was performed with Welch's two-tailed $t$ test. Statistical analysis to compare the observed frequencies of two categories was performed with two-sided binomial test of the null hypothesis that the probability of success in a Bernoulli experiment is 0.5. $P < 0.05$ was considered as statistically significant. Values are represented as means ± SD as indicated in the figures. Tracking data for GNRs were obtained from at least five replicates in at least two independent experiments, and sample sizes are indicated in detail in the text.

**Reporting summary**. Further information on research design is available in the Nature Portfolio Reporting Summary linked to this article.

## Data availability

The datasets generated and/or analysed and all samples used in this study are available from the corresponding author on reasonable request. Source data for figures can be found in Supplementary Data 1.

## Code availability

Custom scripts were written for tracking GNR and for analysis of trajectories and intensities with LabVIEW. These scripts are available on reasonable request. LabVIEW source code and executable software for the Monte Carlo simulations used in this study are also available from GitHub (https://github.com/MitsuSGW/MonteCarlo-MotorTeam-Sim.git).

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

## Acknowledgements

CBD-TEV-ZEN4/pGEX-6p plasmid was a gift from Prof. Masanori Mishima. We thank Prof. TOYOSHIMA Y. Yoko and SAITO Kei for critical discussion; Adam Brotchie, PhD, from Edanz Group (www.edanzediting.com/ac) for editing a draft of this manuscript. This work is supported by Ministry of Education, Culture, Sports, Science and Technology Japan (MEXT) KAKENHI Grant Numbers JP19K06593 and JP19H03190 (to M.S.), JP20K06635 and JP21K19252 (to J.Y.): MEXT KAKENHI Grant-in-Aid for Scientific Research on Innovative Areas Grant Number JP21H00386 and for Transformative research Areas Grant Number JP21H05868 (to J.Y.): Research Foundation for Opto-Science and Technology Japan (to M.S.).

## Author contributions

M.S. and J.Y. designed research; M.S. performed experiments, analyzed data, and performed simulations; Y.M., and M.Y. contributed reagents; M.S., R.A.C. and J.Y. wrote the paper.

## Competing interests

The authors declare no competing interests.
