## [Peer Review File · Communications Biology]

Reviewers' comments:

Reviewer #1 (Remarks to the Author):

Sugawa et al. describe a polarization microscopy-based assay that enables them to measure that purified non-processive or weakly processive molecular motors of the kinesin family can carry cargo along microtubules in a helical fashion driven by the rotational component of the force generated by each motor of a team of a hundred or so attached to gold nanorods as cargo. This helix has a much smaller pitch than the microtubule supertwist and is of the order of a micron. The measurements allow the authors to deduce that accompanied with and as a condition for this helical motion is a rolling motion of the gold nanorods as well as yaw around the nanorod's short axis. This helical transport is observed for a small fraction of the sampled nanorods, and is attributed to the relative orientation of the nanorods and the microtubules.

The assay convincingly confirms that kinesin motors rotate with every step as has been previously demonstrated for kinesins in the references cited in the manuscript. The nanorods themselves constitute a very special motor-cargo geometry, and the authors do not comment on where in the cellular context such a phenomenon is expected to be observed, or be relevant. While the choice of the nanorods was obviously made to allow the polarization measurements, it is suggested in the manuscript that similar helical motion should ensue with cellular cargo. A main difference between cellular cargo and the nanorods is the way the motors are bound to the cargo. The motors were rigidly attached to the nanorods through specific linkers while in cells, motors are bound to the fluid membrane surfaces of vesicles and organelles and are therefore typically free to diffuse within the membrane. It is not clear how torque would be transmitted to the cargo in this case, whether cargoes will similarly move in a helical fashion, and whether the motor forces can be transmitted to the cargo to force it to yaw (spin in the case of spherical cargo). It is also not clear if the density of 100 or so motors on a ~50nm nanorod captures the motor density on organelles or vesicles in cells. The suggestion based on the simulation that the observed helical motion could help cellular cargo avoid obstacles, will require that cellular cargo move in the fashion described for nanorods.

additional comments:

1) The authors estimate a density of 100 motors per nanorod. Examples of cellular cargo with such a density would be important to mention in the manuscript. How many of these motors are expected to be bound to the microtubule at any point of time? What does changing the density of the motors do to this helical mode of motion?

2) Related to this, the authors assert that nanorods with a larger aspect ratio than the ones they used would not be expected to exhibit the motion they describe since they align with the microtubule. This should be easy to demonstrate given the availability of nanorods with various aspect ratios from the same supplier the authors use. How would lowering the motor density per nanorod alter the motion of high aspect ratio cargoes?

3) The mathematical model briefly described in the methods is not clear. What analysis results in the conclusion that roll frequency is twice the yaw frequency? This needs to be further clarified.

4) In that section of the methods there is a mention of sK1-GNR experiments. I am not sure those are described in the manuscript.

5) In relation to figure 6 and the associated video, one would expect that the microtubule advances a few microns as it rotates many times, but the video and images clearly show that it does not. Why is that?

6) The authors use simulation to show that the mode of motion they describe increases the chances of the cargo avoiding obstacles. Could they use a simple in vitro assay and the nanorods to demonstrate this? Potentially by specifically and firmly attaching obstacles to the microtubule – e.g. through an anti-tubulin bridge.

Reviewer #2 (Remarks to the Author):

In the manuscript "Torque component in the kinesin stroke drives coupled yawing and orbital rotations of cargo", Sugawa et al developed a gold nanorod (GNR) based assay that enables the investigation of the dynamics of cargo transported along the microtubule (MT) by a team of kinesins. On the basis of their analysis, the authors show that, besides rotating around the MT, the GNR yaws and rolls, and using a Monte Carlo model they argue that this movement enables the kinesin team to overcome obstacles.

The questions addressed by the authors are interesting and difficult to tackle. The experimental methodology introduced by the authors holds the promise to be helpful, however there are a few issues. First, it seems that the observations made by the authors are confined to GNRs whose size is restricted to the one considered by the authors. Even though this might still be informative, it raises the question as to whether the findings apply to in vivo conditions as well. More importantly, I have some reservations pertaining to the Monte Carlo model, and the equations used to interpret the data.

Major Point: The mathematical formulation of the GNR trajectories should be revisited. To see where the problem is, note that in equation (1) the matrices for roll and yaw do not commute, and thus I suspect that equation (2) would be different if the equation had been written as $R_{yaw}R_{roll}0$. Because these equations are central in the analysis performed by the authors, either the authors can justify them on solid grounds, or perhaps they should resort to a classical mechanics formulation of the rigid body rotation.

Major Point: From authors comments (lines 79-82 and lines 268-279) the analysis made in the manuscript pertains only to GNRs that are small enough so that they are not aligned to the MT, and yet large enough so that the fluctuations described are stable. For the GNRs in the experiment, only 20% of the time these stable rotations are observed. Have the authors attempted to repeat the measurements using a GNR of different size to test their hypotheses? Would these rotations occur in the typical configuration of a team of kinesins transporting cargo in vivo?

Major Point: The Monte Carlo simulations have a few issues. First of all, the method is not clearly explained. The authors should provide an algorithm that illustrates step-by-step how the system is evolved in time. Second, particles (kinesins) and the rigid body (GNR) are subject to thermal motion, which is not accounted for here. Third, at every step the location and orientation of cargo are chosen so to fulfill force and torque balance equations. However, it seems as if the rotation due to the stepping particles is applied (lines 574-575) after force and momentum conservation are satisfied. If this is the case, the rationale is not clear. Fourth, satisfying force and torque balance equations does not preclude the possibility that the system attains conformations of very high energy, with unrealistic stretch of the springs. This could be resolved by including a force-dependent dissociation constant of kinesin from the MT. Fifth, despite the simplicity of the model, it requires many parameters and it is not clear of much they impact the final result. Overall, the authors should improve their model, or possibly remove it given that it is not the main focus of the work.

Minor Point: Does the roll imply that the same side of the GNR faces the MT throughout the trajectory? Is this to be expected? If not, is it possible that inhomogeneous distributions of motors around the GNR impact the trajectory of the GNR?

Minor Point: Line 62: kinesin-1 is missing.

Our response to the reviewers' comments

Original Reviewers' comments

Our response

Reviewer #1 (Remarks to the Author):

The nanorods themselves constitute a very special motor-cargo geometry, and the authors do not comment on where in the cellular context such a phenomenon is expected to be observed, or be relevant. While the choice of the nanorods was obviously made to allow the polarization measurements, it is suggested in the manuscript that similar helical motion should ensue with cellular cargo. A main difference between cellular cargo and the nanorods is the way the motors are bound to the cargo. The motors were rigidly attached to the nanorods through specific linkers while in cells, motors are bound to the fluid membrane surfaces of vesicles and organelles and are therefore typically free to diffuse within the membrane. It is not clear how torque would be transmitted to the cargo in this case, whether cargoes will similarly move in a helical fashion, and whether the motor forces can be transmitted to the cargo to force it to yaw (spin in the case of spherical cargo). It is also not clear if the density of 100 or so motors on a ~50 nm nanorod captures the motor density on organelles or vesicles in cells. The suggestion based on the simulation that the observed helical motion could help cellular cargo avoid obstacles, will require that cellular cargo move in the fashion described for nanorods.

We much appreciate this reviewer for his/her careful reading and constructive criticism of our paper. Indeed, in our original submission we did not comment on the potential biological relevance of our findings, or on the generalizability of our findings. In our revised manuscripts, we address these and the reviewer's other points, as detailed point by point below.

Biological relevance. The reviewer is of course correct that kinesins linked to an adaptor floating in the lipid membrane of a cargo vesicle will be free to rotate so as to dissipate torque. Crucially however, and we apologise for not making this explicit, there are other cellular cargoes to which kinesins attach rigidly, for example IFT particles, viruses, ribonuclear proteins, transport microtubules, actin filaments and possibly endosomes (which have an external skeleton of clathrin triskelia). We now clarify that we envisage that our data will be relevant to situations where teams of non-processive kinesins are linked to these types of solid cargoes, rather than to lipid bilayer enveloped cargoes (page 10, lines 276-284).

additional comments:

1) The authors estimate a density of 100 motors per nanorod. Examples of cellular cargo with such a density would be important to mention in the manuscript. How many of these motors are expected to be bound to the microtubule at any point of time? What does changing the density of the motors do to this helical mode of motion?

We sought to work always with a close packed team of surface-mounted kinesins. As we originally discussed, this corresponds to about 100 molecules per GNR, but only a small fraction of these will interact with the microtubule at any one time. Simple geometrical considerations suggest that kinesins could only bridge between the two surfaces for around 20% of the microtubule circumference and around 20% of the GNR circumference. In addition, the low duty ratio of non-processive kinesins implies that only a small fraction of kinesins will be strongly attached and making force at any one moment. As we now discuss (page 4, lines 88-90) if we assume the duty ratio is about 20%, then perhaps 5-10 kinesin motor domains might generate force at any one time. Molecular teamwork is a requirement for non-processive kinesins to move cargo.

2) Related to this, the authors assert that nanorods with a larger aspect ratio than the ones they used would not be expected to exhibit the motion they describe since they align with the microtubule. This should be easy to demonstrate given the availability of nanorods with various aspect ratios from the same supplier the authors use. How would lowering the motor density per nanorod alter the motion of high aspect ratio cargos?

We now clarify that whilst, as the reviewer notes, we chose these slightly asymmetrical nanoparticles in order to obtain an orientation signal, we expect our results to relate to any symmetrical or moderately symmetrical molecular cargo to which kinesins are rigidly attached. We apologise for not making this clearer. Kinesin-coated cargoes with a very high aspect ratio are known to align their long axes to that of the track microtubule. As we discuss, a clear example is Ncd-driven microtubule sliding, where the long axis of the transport microtubule aligns to that of the track microtubule. The short microtubules (1~2 μm), of which the aspect ratio was 40~80, stably moved in helical fashion on the longer microtubule, not showing the yawing rotation (page 4, line 84, reference #16, Mitra *et al.*, *Nature Communications*. 2020;11(1):2565). We envisage that torque force is still developed by kinesin teams that transport high aspect ratio cargoes, but that torque is opposed by a tendency to maximise the kinesin-containing interface between the two microtubules. From our earlier work on kinesin-driven rolling of sliding microtubules, we know that rolling can be blocked without affecting on-axis progress (reference #17, Yajima & Cross. *Nature Chemical Biology* 2005;1:338-341).

The reviewer is right that in principle we could define the cutoff value for the aspect ratio at which oblate kinesin-cargoes switch from rolling and yawing (low aspect ratio) to rolling only (high aspect ratio). However, this would require repeating our whole study multiple times and acquiring a new laser with a different wavelength (see table below). We hope to pursue this interesting question in the future but at present our resources are insufficient to do this. Our goal with the current manuscript is to evidence rolling and tumbling of cargoes with low aspect ratio.

3) *The mathematical model briefly described in the methods is not clear.*

Thank you for this comment. We added a detailed description of the model.

Pages 17-18, lines 520-551

Mathematical model of helical and rotational motion of a kinesin-coated GNR on a microtubule. The GNR motility assay revealed that KIF1A- and ZEN4-GNRs unidirectionally rotate about the GNR's short axis, moving along the left-handed helix track. The helix track of the center position of a GNR is given by

$$helix(t, v, \omega_{Helix}) = \begin{pmatrix} X(t) \\ Y(t) \\ Z(t) \end{pmatrix} = \begin{pmatrix} vt \\ h \cdot \cos(\pi/2 - \omega_{Helix}t) \\ h \cdot \sin(\pi/2 - \omega_{Helix}t) \end{pmatrix}$$

(8)

where v is the translational velocity along the X-axis, ω_{Helix} is the angular velocity in the Y-Z-plane, and h is the radius of the helix. Assuming that the long axis of the GNR is parallel to the X-Y-plane under the initial conditions, the initial position matrix p_0 consisting of the tip, center, and end positions of the GNR is given by

$$p_0(\text{tip, center, end}) = \begin{pmatrix} r \cdot \cos(\varphi) & 0 & r \cdot \cos(\varphi + \pi) \\ r \cdot \sin(\varphi) & 0 & r \cdot \sin(\varphi + \pi) \\ 0 & 0 & 0 \end{pmatrix} \quad (9)$$

where r is the radius along the long axis of the GNR and φ is its phase angle. The experimental results suggest that a kinesin-coated GNR unidirectionally rotates about its short axis and further around the microtubule, so that the rotation comprises yawing and rolling. The elemental rotation matrices are thus given by

$$R_Z(\omega_{Yaw}t) = \begin{pmatrix} \cos(\omega_{Yaw}t) & -\sin(\omega_{Yaw}t) & 0 \\ \sin(\omega_{Yaw}t) & \cos(\omega_{Yaw}t) & 0 \\ 0 & 0 & 1 \end{pmatrix} \quad (10)$$

$$R_Y(0) = \begin{pmatrix} 1 & 0 & 0 \\ 0 & 1 & 0 \\ 0 & 0 & 1 \end{pmatrix} \quad (11)$$

$$R_X(\omega_{Roll}t) = \begin{pmatrix} 1 & 0 & 0 \\ 0 & \cos(\omega_{Roll}t) & -\sin(\omega_{Roll}t) \\ 0 & \sin(\omega_{Roll}t) & \cos(\omega_{Roll}t) \end{pmatrix} \quad (12)$$

where ω_{Yaw} and ω_{Roll} are the angular velocities of yawing and rolling, respectively. The rotation matrix of the extrinsic element rotation sequence for our proposed motion is given by $R_X R_Y R_Z$. The helical and rotational trajectory of a motor-coated GNR is thus given as Equation (1). By calculating Equation (1),

$$\text{tip}(t) = \begin{pmatrix} X_{\text{tip}}(t) \\ Y_{\text{tip}}(t) \\ Z_{\text{tip}}(t) \end{pmatrix} = \begin{pmatrix} r \cdot \cos(\omega_{Yaw}t + \varphi) + vt \\ r \cdot \cos(\omega_{Roll}t) \cdot \sin(\omega_{Yaw}t + \varphi) + h \cdot \cos(\pi/2 - \omega_{Helix}t) \\ r \cdot \sin(\omega_{Roll}t) \cdot \sin(\omega_{Yaw}t + \varphi) + h \cdot \sin(\pi/2 - \omega_{Helix}t) \end{pmatrix}$$

$$\text{center}(t) = \begin{pmatrix} X_{\text{center}}(t) \\ Y_{\text{center}}(t) \\ Z_{\text{center}}(t) \end{pmatrix} = \begin{pmatrix} vt \\ h \cdot \cos(\pi/2 - \omega_{\text{Helix}}t) \\ h \cdot \sin(\pi/2 - \omega_{\text{Helix}}t) \end{pmatrix}$$

$$\text{end}(t) = \begin{pmatrix} X_{\text{end}}(t) \\ Y_{\text{end}}(t) \\ Z_{\text{end}}(t) \end{pmatrix} = \begin{pmatrix} -r \cdot \cos(\omega_{\text{Yaw}}t + \varphi) + vt \\ -r \cdot \cos(\omega_{\text{Roll}}t) \cdot \sin(\omega_{\text{Yaw}}t + \varphi) + h \cdot \cos(\pi/2 - \omega_{\text{Helix}}t) \\ -r \cdot \sin(\omega_{\text{Roll}}t) \cdot \sin(\omega_{\text{Yaw}}t + \varphi) + h \cdot \sin(\pi/2 - \omega_{\text{Helix}}t) \end{pmatrix}.$$

Therefore, the time trajectory of the GNR orientation in the X-Y-plane is given by

$$\frac{Y_{\text{tip}}(t) - Y_{\text{end}}(t)}{X_{\text{tip}}(t) - X_{\text{end}}(t)}$$

and finally Equation (2) is obtained. To synchronize the rolling and the helix truck, the angular velocity of the rolling (ω_{Roll}) is identical to that of the helix truck (ω_{Helix}). And the angular velocity of the yawing (ω_{Yaw}) is half that of the helix track because the GNR rotates half during one period of the helix truck. The calculated helical and rotational trajectories of a kinesin-coated GNR and the time trajectories of the GNR orientation with different phase angles are shown in Fig. 4, Fig. S6, and Movies S2, S4, and S5.

3) What analysis results in the conclusion that roll frequency is twice the yaw frequency? This needs to be further clarified.

Based on figure 3C (for KIF1A-GNR) and figure 5B (for ZEN4-GNR), which showed a linear relationship between polarization pitch (from 0 degree to 180 degree) and helical pitch (proportionality factor, approximately 1), we estimated that one full yawing motion is twice the rolling motion. To clarify this point, we now revised the sentences of our description.

Page 6, lines 149-168

Using the extrinsic rotations about the Z-, Y-, X-axes in that order, our proposed motion of the kinesin-coated GNR is given by

$$R_X(\omega_{\text{Roll}}t)R_Y(0)R_Z(\omega_{\text{Yaw}}t)p_0(\varphi) + \text{helix}(t, v, \omega_{\text{Helix}}), \quad (1)$$

where $R_X(\omega_{\text{Roll}}t)$ is the rolling matrix with the angular velocity ω_{Roll} , $R_Y(0)$ the pitching matrix with zero angle, $R_Z(\omega_{\text{Yaw}}t)$ the yawing matrix with the angular velocity ω_{Yaw} , p_0 an initial position of a kinesin-GNR with an initial phase angle φ , and $\text{helix}(t, v, \omega_{\text{Helix}})$ a helical displacement with translational velocity v and angular velocity ω_{Helix} as functions of time t (Materials and Methods). The initial phase angle φ is defined by the relative angle to the microtubule axis when the kinesin-coated GNR is at the top position of its helical trajectory. The polarization angle measured in this study was the orientation of the GNR projected in the X-Y-plane. Thus, time trajectories of the GNR angle in the X-Y-plane are given by

$$\arctan[\cos(\omega_{\text{Roll}}t) \cdot \tan(\omega_{\text{Yaw}}t + \varphi)] \quad (2)$$

(Materials and Methods). To synchronize the rolling and the helix truck, the angular velocity of the rolling (ω_{Roll}) is identical to that of the helix truck (ω_{Helix}). And the angular velocity of the yawing (ω_{Yaw}) is half that of the helix track because the GNR rotates 180° during one period of the helix truck (Fig. 2D and Fig. 3A). Therefore, the time trajectories of the KIF1A-GNR angle were well explained by this model at $\omega_{\text{Helix}} = \omega_{\text{Roll}} = \pm 2\omega_{\text{Yaw}}$ (Fig. 4B). This means that a KIF1A-coated GNR unidirectionally rotates 180° about its short (yaw) axis in one period of short-pitch helical motion around the microtubule long axis, which is consistent with our observation that one full yawing motion is twice the rolling (helical) motion (Fig. 3C).

4) In that section of the methods there is a mention of sKI-GNR experiments. I am not sure those are described in the manuscript.

Thank you very much for careful reading. We are sorry to confuse you. We revised this sentence (page 13, line 398).

5) In relation to figure 6 and the associated video, one would expect that the microtubule advances a few microns as it rotates many times, but the video and images clearly show that it does not. Why is that?

Dimeric ZEN-4 tends to accumulate at the plus end of microtubules (Hutterer *et al.*, Current Biology (2009) 19:2043-2049). Therefore, the short microtubule continuously rotated with the tip attached to ZEN4-GNR. We added this point in the main text.

Page 8, lines 212-213

Due to the property of ZEN4 dimer to accumulate at microtubule plus ends (26), these short microtubules continuously rotated with the microtubule tip attached to the ZEN4-GNR.

6) The authors use simulation to show that the mode of motion they describe increases the chances of the cargo avoiding obstacles. Could they use a simple in vitro assay and the nanorods to demonstrate this? Potentially by specifically and firmly attaching obstacles to the microtubule – e.g. through an anti-tubulin bridge.

We accept the reviewer's point that the possible role for yaw axis rotation in avoidance of obstacles would be better addressed by experiment rather than simulation. In future work, we hope to address this point, though we will need an obstacle larger than an antibody. For the present

we accept that our simulation required too many potentially unrealistic assumptions and we have removed it and associated discussion from the manuscript.

Reviewer #2 (Remarks to the Author):

Major Point:

The mathematical formulation of the GNR trajectories should be revisited. To see where the problem is, note that in equation (1) the matrices for roll and yaw do not commute, and thus I suspect that equation (2) would be different if the equation had been written as $R_{yaw}R_{roll}p_0$. Because these equations are central in the analysis performed by the authors, either the authors can justify them on solid grounds, or perhaps they should resort to a classical mechanics formulation of the rigid body rotation.

We appreciate this reviewers' critical constructive comments on our proposed model. We understand his/her concern about the equation (2). The sequence of the rotation matrices is critical to describe the orientation of a rigid body with respect to a fixed coordinate system (please see, for example, "Euler angles" on Wikipedia). Indeed the matrices for roll and yaw do not commute, the proposed rotational motion is only given by $R_{yaw}R_{roll}p_0$. To clarify this point, we now added more detailed description in Materials and Methods (page 17, lines 520-551), as mentioned in our response to the reviewer #1 additional point 3.

Major Point:

Have the authors attempted to repeat the measurements using a GNR of different size to test their hypotheses?

Please see our response to reviewer 1 comment #1.

Would these rotations occur in the typical configuration of a team of kinesins transporting cargo in vivo?

Please see our response to reviewer 1 *Biological relevance*.

Major Point: The Monte Carlo simulations have a few issues. First of all, the method is not clearly explained. The authors should provide an algorithm that illustrates step-by-step how the system is evolved in time.

We really appreciate this reviewer's constructive criticism and suggestions. We have now added the pseudo code of the Monte Carlo simulation in Supplementary information text (pages 2-4).

Second, particles (kinesins) and the rigid body (GNR) are subject to thermal motion, which is not accounted for here.

Our simulation is not based on the Langevin dynamics. The main purpose of the simulation is to verify the suggestion obtained by the ratchet model and not to create the realistic motion model so that our simple Monte Carlo simulation is enough for this purpose.

Third, at every step the location and orientation of cargo are chosen so to fulfill force and torque balance equations. However, it seems as if the rotation due to the stepping particles is applied (lines 574-575) after force and momentum conservation are satisfied. If this is the case, the rationale is not clear.

In the simulation, we assumed that the stepped particles further generate torque to rotate the cargo, in addition to the moment generated by stretching the spring. In response to this important comment, we carefully reconsidered our rationale and as a result slightly modified the calculation of the moment as shown below (page 19, line 578).

$$\sum_i^{N_{\text{on}}} k \vec{d}_i \times \vec{l}_i - T_{\text{step}} N_{\text{step}} = 0$$

where k is the spring constant; \vec{d}_i is an extension vector of the spring of the i -th particle that is bound to the rail; \vec{l}_i is the position vector of the i -th joint relative to the center of the joint points connecting the particles that are bound to the rail; and N_{on} is the number of the particles bound to the rail; T_{step} is the torque per step of single motor particle; N_{step} is the number of motor particles taking a step in the unit time. An orientation of the cargo is determined to minimize the value of the left side of the equation. We added the pseudo code of the algorithm in Supplementary information text. We revised our description of the Monte Carlo simulation method and results (lines 250-263 in the main text, lines 572-594 in the Materials and Methods section, Fig.7E and F, Supplementary information text (pages 2-4)). Thank you for this constructive criticism. These changes improve clarity and improve the accuracy of the simulation, whilst our conclusion remains the same as in the previous version.

Page 9, lines 250-263

To test this suggestion, we performed simple 2D Monte Carlo simulations. Multiple motor particles are linked to a rigid body cargo with a spring, stochastically taking biased

forward and/or sideward steps on the 2D lattice, driving force and moment applied to the cargo, and generating additional torque (T_{step}) to rotate the cargo when taking a step (Fig. 7E). Displacement and rotation of the cargo were based on the force- and moment-balance equations (Materials and Methods). When T_{step} was -24 , the cargo exhibited periodical clockwise rotation (Fig. 7F left). The rotation period of the cargo roughly matched the 13-lattice period of the Y-displacements modeled as the 13-protofilament microtubule (Fig. S7). As the theory predicted, the lateral displacement of the cargo with torque generation increased compared to that without torque generation (Fig. 7F right). In the experiments as shown above, the pitches of the helical motion of KIF1A-GNR ($\sim 0.7 \mu\text{m}$) and ZEN4-GNR ($\sim 0.7 \mu\text{m}$) were shorter than those of the corkscrewing of microtubules on the surfaces of KIF1A ($\sim 0.9 \mu\text{m}$) and ZEN4 ($\sim 0.8 \mu\text{m}$ (11)), respectively. These results support the analysis by the noise-driven ratchet model showing that unidirectional rotation of a motor-coated cargo enhances the sideward bindings of motor molecules toward the biased direction.

Page 19, lines 581-591

T_{step} is the torque per step of single motor particle; N_{step} is the number of motor particles taking a step in the unit time.

The simulation was developed in LabVIEW (National Instruments). Pseudo code of the simulation is shown in Supplementary information. LabVIEW source code and executable software for the Monte Carlo simulations used in this study are also available from GitHub (<https://github.com/MitsuSGW/MonteCarlo-MotorTeam-Sim.git>). The values for the parameters used in the simulation were $k=1$; $\tau_{\text{on}} = 40$ units of time; $\tau_{\text{off}} = 10$ units of time; $\alpha_x = 0.12$ units of lattice size; $\alpha_y = 0.58$ units of lattice size; $\sigma = 0.11$ units of lattice size; $T_{\text{step}} = 0$ or -24 . The parameters for the biased forward and leftward displacements (τ_{on} , τ_{off} , α_x , α_y , σ) were optimized so that the ratio of forward to leftward steps is approximately 100:13 that is equivalent to the 0.8- μm helix pitch of KIF1A and ZEN-4 in the gliding assay (Fig. 7F).

Fourth, satisfying force and torque balance equations does not preclude the possibility that the system attains conformations of very high energy, with unrealistic stretch of the springs. This could be resolved by including a force-dependent dissociation constant of kinesin from the MT.

The algorithm of stepping of particles is given by

1. **INPUT** particle state $(x_i^p, y_i^p, state_i^p, dwell_i^p)$
2. **IF** $x_i^p \bmod (1) > \alpha_x$ **THEN** $x_i^p = \text{integer}(x_i^p) + 1$

3. **ELSE THEN** $x_i^p = \text{integer}(x_i^p)$
4. **END IF**
5. **IF** $y_i^p \bmod (1) > \alpha_y$ **THEN** $y_i^p = \text{integer}(y_i^p) + 1$
6. **ELSE THEN** $y_i^p = \text{integer}(y_i^p)$
7. **END IF**
8. **RETURN** particle state $(x_i^p, y_i^p, \text{state}_i^p, \text{dwell}_i^p)$

Positions of particles (x_i^p, y_i^p) fluctuated based on Gaussian with standard deviation of $\sigma = 0.11$. And the asymmetric factors α_x and α_y were set to 0.12 and 0.58, respectively. These parameters were optimized to match the experimental results as we mentioned in Materials and Methods. For example, the x -displacement histogram obtained by the simulation in which the start position was set to 0 indicates that the unrealistic stretch of the spring could not happen in our simulation. Note that the probability of forward step is 0.14 in this case. Therefore, the problem that the reviewer#1 was concerned about does not occur in our simulation.

Fifth, despite the simplicity of the model, it requires many parameters and it is not clear of much they impact the final result. Overall, the authors should improv their model, or possibly remove it given that it is not the main focus of the work.

The main purpose of the simulation is to verify the analyses of the ratchet model that the yawing rotation facilitates the biased lateral movement. As the reviewer suggested, we revised the simulation results (page 9, lines 250-263 and page 19, lines 571-591 in the main text, Fig.7E-F, and Supplementary information text).

Minor Point: Does the roll imply that the same side of the GNR faces the MT throughout the trajectory? Is this to be expected? If not, is it possible that inhomogeneous distributions of motors around the GNR impact the trajectory of the GNR?

We envision that the GNR rolls as it glides over the MT surface, driven by the same forces that cause microtubules to roll as they slide on coverslips coated in non-processive kinesins. This

rolling has no impact on the instantaneous forces because we assume a uniform density of motors on the GNR. To achieve this, we sought to saturate kinesin binding to the GNR. Indeed inhomogeneous distributions of motors around the GNR might impact the trajectory of the GNR, because the amount of torque would vary. In all situations where a team of weakly processive kinesins is engaged, torque will be produced in proportion to the number of engaged motors.

Minor Point: Line 62: kinesin-1 is missing.

We have corrected this sentence.

Page 3, lines 64-65.

We performed the GNR motility assay on the plus-end-directed kinesins KIF1A, which is monomeric, and ZEN-4, which is a weakly processive dimer.

Reviewers' comments:

Reviewer #2 (Remarks to the Author):

The authors reply clarified many of the points that I have raised. However, there are still a couple of issues that I think should be resolved in order to clarify the analysis.

(1) Minor Point: Ratchet Model (Eqs. 3-4, 12-14): There are some units discrepancies (for instance with the unitless a and the length d , and the ratio in the argument of the exponential in Eq. 12). Some of them can be solved by introducing the diffusion coefficient used in the model, or some step size associated with the displacement of kinesin. Even though within the model they are set to 1, this should be clearly stated, and they should appear in the equations (at least once) for clarity. In addition, I am confused by the fact that the authors first estimate the value of the off-axis asymmetry factor (0.2) and then use a different one taken from the literature (0.57).

(2) Minor Point: Monte Carlo (MC) Simulations (Main Text. lines 571-593, and Supporting Information, pages 2-4). The changes made by the reviewers make the procedure more clear. There are still some issues that need clarification.

(a) In the algorithm to compute the force and moment balance, the new cargo state is determined using the old location of the hinges, which are then updated after the new cargo state is produced. While the algorithm checks for convergence of the moment, is the force-balance equality fulfilled as well? I suspect that this might be the case after a few iterations, but a control would be appropriate. Furthermore, is the convergence parameter 0.5 strict enough?

(b) What justifies the choice of the value given to T_{step} ?

(c) In replying to my point about conformations of high energy, the authors have checked that individual steps are not too large. To my understanding, that is regulated by the diffusion of the unbound head alone. What should be controlled, I think, is the distribution of the distance between particle position and the joint, which could be unrealistically wide, even though the mean is zero because of the force constraint.

(d) If I understood correctly, in the routine "calculate new binding site", the state should be updated as well.

(3) Minor Point: The authors mention that rolling and helical movement are synchronized (lines 161 ad 546). Although this makes sense intuitively, I think that the authors should expand on this. For instance, is it possible that the cargo does not roll at all given that it is uniformly covered by kinesins? This goes back to the previous point that I raised, regarding whether always the same side of the GNR is closest to the MT.

Reviewers' comments:

Reviewer #2 (Remarks to the Author):

The authors reply clarified many of the points that I have raised. However, there are still a couple of issues that I think should be resolved in order to clarify the analysis.

(1) Minor Point: Ratchet Model (Eqs. 3-4, 12-14): There are some units discrepancies (for instance with the unitless α and the length d , and the ratio in the argument of the exponential in Eq. 12). Some of them can be solved by introducing the diffusion coefficient used in the model, or some step size associated with the displacement of kinesin. Even though within the model they are set to 1, this should be clearly stated, and they should appear in the equations (at least once) for clarity. In addition, I am confused by the fact that the authors first estimate the value of the off-axis asymmetry factor (0.2) and then use a different one taken from the literature (0.57).

Answer. We very much appreciate this reviewer for his/her careful reading and constructive criticism of our paper. The parameter “ d ” is normalized by the off-axis distance between the binding sites and is therefore dimensionless. The value of 0.2 is not the asymmetry factor (α) but represents the normalized lateral displacement of motor particles driven by rotation of the cargo when the length between the binding sites toward the off-axis is set to 1. We have revised our manuscript to clarify these points, changing the sentences in the Results section (Lines 232-234, 236 and 243) and also the “Evaluation of particle flux in the noise-driven ratchet model” section in the Material and Methods (Lines 563-564).

Page 8, lines 232-236

Lateral displacement of 1 nm is equivalent to $0.2 \approx 1 \text{ nm}/5.1 \text{ nm}$ (27) when the length between the binding sites toward the off-axis is set to 1, which would be sufficient to influence the noise-driven ratchet mechanism (Fig. 7b). A recent discussion by Mitra et al. indicates that around 1 μm of a helix pitch is reproduced by the 2D Brownian ratchet model with a slight off-axis asymmetry factor relative to the on-axis asymmetry factor (19).

Page 8, lines 242-243

where J_0 represents the flux of particles starting diffusion at 0 and $J_{\pm d}$ represents the flux of particles starting diffusion at $\pm d$ when the length between the binding sites toward the off-axis is set to 1.

The parameter d is normalized as the length between the binding sites toward the off-axis is set to 1.

(2) Minor Point: Monte Carlo (MC) Simulations (Main Text, lines 571-593, and Supporting Information, pages 2-4). The changes made by the reviewers make the procedure more clear. There are still some issues that need clarification.

(a) In the algorithm to compute the force and moment balance, the new cargo state is determined using the old location of the hinges, which are then updated after the new cargo state is produced. While the algorithm checks for convergence of the moment, is the force-balance equality fulfilled as well? I suspect that this might be the case after a few iterations, but a control would be appropriate. Furthermore, is the convergence parameter 0.5 strict enough?

Answer. The force-balance is calculated in the first step in the following simulation time so that the force-balance equality is continuously fulfilled. The graph below shows the typical time trajectory of the moment before calculating the moment balance in the simulation without the extra torque ($T_{\text{step}} = 0$). The stepping event of motor particles generated impulses of moment reaching to around ± 5 .

Next, the second graph below shows the typical time trajectory of rotation angle per one unit time, which was obtained from the same simulation as shown above. The rotation angle driven by stepping events was typically within ± 1 degree.

Taken together, the convergence factor of 0.5 is determined to be effective. We performed the simulations with lower convergence parameters and found that this did not affect the conclusions.

(b) What justifies the choice of the value given to Tstep?

Answer. The value itself has no physical meaning, but our simulations show that there is a certain magnitude of torque such that the rotation of cargo strongly correlates with the Y-displacement.

(c) In replying to my point about conformations of high energy, the authors have checked that individual steps are not too large. To my understanding, that is regulated by the diffusion of the unbound head alone. What should be controlled, I think, is the distribution of the distance between particle position and the joint, which could be unrealistically wide, even though the mean is zero because of the force constraint.

Answer. The reviewer's point relates to the generalisability of our Monte Carlo simulation. Here, we make no claims to generalizability. Our simulation is intended to throw light specifically on the GNR assay results. Other configurations of particles are not intended to lie within the scope of the simulation. We agree with the idea that the generality of our simulation may be limited to conditions close to the scale ratio of the GNR and the microtubules used in our experiments.

(d) If I understood correctly, in the routine "calculate new binding site", the state should be updated as well.

Answer. The "Calculate new binding site" function only calculates step size of particles in a single unit of time and is a subfunction for the "Calculate next particle states" function. The particle states are updated in the "Calculate next particle states" function.

(3) Minor Point: The authors mention that rolling and helical movement are synchronized (lines 161 ad 546). Although this makes sense intuitively, I think that the authors should expand on this. For instance, is it possible that the cargo does not roll at all given that it is uniformly covered by kinesins? This goes back to the previous point that I raised, regarding whether always the same side of the GNR is closest to the MT.

Answer. This is an important point and we have been encouraged by the reviewer's comments to expand slightly upon it (Lines 317-323). We showed in previous work that while straight microtubules exhibit continuous corkscrew motion in the microtubule gliding motility assay (Ref 17 in the main text), microtubules with a side-extension are prevented from rolling, yet slide at a very similar velocity to rolling microtubules. These experiments suggest that indeed translocation of a kinesin-coated cargo without rolling may be possible. We assume that this rarely occurs in the case of the GNR, because the helical orbital motion of the GNR around the microtubule is clear, and in the absence of any constraint, rolling of the GNR would be expected in reaction to the same forces. Rolling of the GNRs is not directly detected by our polarization measurements. We are much interested in this point, but further study will be needed to verify whether kinesin-GNRs roll during the helical and yawing motion.

Page 11, lines 317-323

We envision that the GNR rolls as it glides over the microtubule surface, driven by the same forces that cause microtubules to roll as they slide on coverslips coated in non-processive kinesins. This rolling has no impact on the instantaneous forces because we assume a uniform density of motors on the GNR. To achieve this, we sought to saturate kinesin binding to the GNR. Indeed inhomogeneous distributions of motors around the GNR might impact the trajectory of the GNR, because the amount of torque would vary. In all situations where a team of weakly processive kinesins is engaged, torque will be produced in proportion to the number of engaged motors.

Reviewer #2 (Remarks to the Author):

I have a couple of comments in response to the authors.

First, explicitly introducing the diffusion coefficient (or equivalently properly defining units of time and space) in equations 3,4,17,18,19 would clarify the physical meaning of $1/\gamma$ time units, and would make the equations more understandable. Right now, there appear to be quantities with units $1/\text{time}^{(1/2)}$ in the argument of the error function, which cannot be. I think that a reader would be aided by correcting this issue, but I also understand that it is a detail and would not affect the point made by the authors.

Second, the configurations of the motors in the Monte Carlo simulations is of interest to understand the features of the model. However, as the authors suggest, it might be beyond the scope of the model.

Reviewer #2 (Remarks to the Author):

First, explicitly introducing the diffusion coefficient (or equivalently properly defining units of time and space) in equations 3,4,17,18,19 would clarify the physical meaning of $1/\gamma$ time units, and would make the equations more understandable. Right now, there appear to be quantities with units $1/\text{time}^{(1/2)}$ in the argument of the error function, which cannot be. I think that a reader would be aided by correcting this issue, but I also understand that it is a detail and would not affect the point made by the authors.

Answer. We very much appreciate this reviewer for his/her careful reading and constructive criticism of our paper. The physical meaning of $1/\gamma$ time units is the time constant at which the velocity approaches the terminal velocity depends on the damping γ (we assume that the diffusing particle has mass 1). We added the explanation about the physical meaning of $1/\gamma$ time units in the Materials and Methods (Line 577-579).

Second, the configurations of the motors in the Monte Carlo simulations is of interest to understand the features of the model. However, as the authors suggest, it might be beyond the scope of the model.

Answer. We very much appreciate this reviewer for the special interests on our Monte Carlo simulation. Since we sought to saturate kinesin binding to the GNR in our experimental condition, other configurations of particles are beyond the scope of this paper.